# NEDD4 controls spermatogonial stem cell homeostasis and stress response by regulating messenger ribonucleoprotein complexes

Zhi Zhou[1,2], Hiroshi Kawabe[3], Atsushi Suzuki[4], Kaori Shinmyozu[5] & Yumiko Saga[1,6,7]

P bodies (PBs) and stress granules (SGs) are conserved cytoplasmic aggregates of cellular messenger ribonucleoprotein complexes (mRNPs) that are implicated in mRNA metabolism and play crucial roles in adult stem cell homeostasis and stress responses. However, the mechanisms underlying the dynamics of mRNP granules are poorly understood. Here, we report NEDD4, an E3 ubiquitin ligase, as a key regulator of mRNP dynamics that controls the size of the spermatogonial progenitor cell (SPC) pool. We find that NEDD4 targets an RNA-binding protein, NANOS2, in spermatogonia to destabilize it, leading to cell differentiation. In addition, NEDD4 is required for SG clearance. NEDD4 targets SGs and facilitates their rapid clearance through the endosomal–lysosomal pathway during the recovery period. Therefore, NEDD4 controls the turnover of mRNP components and inhibits pathological SG accumulation. Accordingly, we propose that a NEDD4-mediated mechanism regulates mRNP dynamics, and facilitates SPC homeostasis and viability under normal and stress conditions.

[1] Division of Mammalian Development, Genetic Strains Research Center, National Institute of Genetics, Yata 1111, Mishima, Shizuoka 411-8540, Japan. [2] School of Life Science and Technology, ShanghaiTech University, Shanghai 201210, China. [3] Department of Molecular Neurobiology, Max Planck Institute of Experimental Medicine, Hermann-Rein Strasse 3D, 37075 Göttingen, Germany. [4] Faculty of Engineering, Division of Materials Science and Chemical Engineering, Yokohama National University, Yokohama, Kanagawa 240-8501, Japan. [5] National Cerebral and Cardiovascular Center, 5-7-1 Fujishiro-dai, Suita, Osaka 565-8565, Japan. [6] Department of Genetics, SOKENDAI, Yata 1111, Mishima, Shizuoka 411-8540, Japan. [7] Department of Biological Sciences, Graduate School of Science, The University of Tokyo, 7-3-1 Hongo, Bunkyo-ku, Tokyo 113-0033, Japan. Correspondence and requests for materials should be addressed to Y.S. (email: ysaga@nig.ac.jp).

Post-transcriptional regulation of messenger RNA (mRNA) translation, sequestration and degradation plays a crucial role in modulating appropriate spatiotemporal gene expression. This RNA-based regulation influences many biological processes, including stem cell homeostasis, embryogenesis and stress response[1–3]. The post-transcriptional regulation of mRNA is controlled by a complicated repertoire of messenger ribonucleoprotein (mRNP) complexes[4]. Therefore, research into post-transcriptional mechanisms of mRNP dynamics in organisms in diverse environments is crucial. These dynamics include mRNP formation and clearance.

P bodies (PBs) and stress granules (SGs) are well-characterized nonmembranous structures storing nontranslated mRNPs in the cytoplasm[5]. PBs usually consist of mRNAs aggregated with mRNA degradation machinery; in general, these mRNPs are present at a low level, but can be upregulated if a large pool of nontranslated mRNAs appears, for example, in spermatogonial stem cells (SSC), satellite cells or neural stem cells[1,3,6]. SGs are aggregates formed under stress conditions such as a low nutrient supply, heat or hypoxia; SGs are thought to represent a pool of mRNPs in a state of translational repression[5]. More recently, SGs have emerged as participants in the pathogenesis of some diseases due to formation of pathological aggregates[7]. Nevertheless, whether these mRNPs are involved in stress response in germ cells is largely unknown.

SSCs retain self-renewal capacity and contribute to the production of spermatozoa throughout the lifetime of a male animal[8]. Undifferentiated spermatogonia are located near the surface of seminiferous tubules that are covered by the basement membrane and peritubular cells. They are classified as $A_{single}$ ($A_s$), $A_{paired}$ ($A_{pr}$) or $A_{aligned}$ ($A_{al}$) spermatogonia according to their morphological features[9]. In adult testes, NANOS2 and GFRα1 are markers of SSCs, which are the most primitive $A_s$ and $A_{pr}$ stem cell populations. $A_{al}$ spermatogonia, marked by NGN3, CDH1 and PLZF, are the transient amplifying spermatogonial progenitor cells (SPCs), which have relatively lower self-renewal ability[10–12]. Entry into differentiation is precisely controlled in response to environmental cues, and downstream signalling events are synchronized with epithelial stages in seminiferous tubules[13]. Sertoli cells secrete glial cell line-derived neurotrophic factor in a stage-dependent manner and promote self-renewal of SSCs (ref. 14). Retinoic acid (RA), another stage-dependent signal, promotes differentiation of germ cells[15]. These environmental signals synchronize differentiation of germ cells and generate a stage-dependent distribution pattern of germ cells[13]. Recently, we provided evidence of a post-transcriptional buffer system controlled by NANOS2-mRNP complexes that protect GFRα1$^+$NANOS2$^+$ stem cells from differentiation signals in the seminiferous tubules[3]. Little is known, however, about the mechanism of removal of this NANOS2-mRNP barrier and the eventual induction of SSC differentiation.

Due to the critical functions of these mRNPs in SSCs and the possible connection to stress-related diseases, it is important to understand the mechanisms that modulate the assembly of PBs and SGs, and their disassembly and clearance from SSCs. One possible regulator is an E3 ubiquitin ligase, NEDD4 (neural precursor cell expressed developmentally downregulated protein 4-1), which is coimmunoprecipitated with NANOS2 in male gonads. Accumulation of the NANOS2 protein promotes mRNP assembly and prevents both proliferation and differentiation of SSCs (refs 3,16), whereas NEDD4 is known to positively regulate cell growth and differentiation in many types of adult stem cells[17,18]. In addition, NEDD4 is the major E3 ligase involved in the clearance of heat-damaged proteins from the cell[19]. Furthermore, deletion of *Itchy*, which encodes another NEDD4 family member, causes germ cell apoptosis and subfertility in male mice[20]. We therefore hypothesized that NEDD4 participates in both mRNP regulation and the heat-stress response during spermatogenesis. To test this hypothesis, we develop systems to genetically manipulate the expression of NEDD4 in male germ cells. We find that the NEDD4 complex regulates mRNP dynamics by targeting mRNP to the lysosomal degradation pathway under normal and heat-stress conditions, thus facilitating differentiation and survival of SPCs under stress.

## Results

**NEDD4 expression in testes under normal and stress states.** NANOS2 plays an essential role in the maintenance of both embryonic male gonocytes and adult SSCs (refs 21,22). In adult testes, NANOS2 and GFRα1 express in $A_s$ and $A_{pr}$ stem cell populations (Fig. 1a). NANOS2 is essential to maintain stem cell state, but the protein has to be cleared for stem cells to enter the differentiation pathway. To elucidate the mechanism of NANOS2 protein regulation, we searched for NANOS2-interacting proteins from male gonadal extracts by immunoprecipitation (IP) followed by mass-spectrometric analysis (Supplementary Fig. 1A). We found NEDD4, an E3 ubiquitin ligase, as a strong candidate of a NANOS2-regulating protein. Given the role of NEDD4 in controlling the fate of adult stem cells in many tissues[17,18], we tested whether NEDD4 also participates in male spermatogonia differentiation. We observed broad expression of NEDD4 in adult testes, but enrichment of NEDD4 was detected in CDH1$^+$ SPCs (Supplementary Fig. 1B,C). In human testes, the NEDD4 protein is detectable in seminiferous tubular germ cells, but its presence is less pronounced in somatic Leydig cells[23]. The conserved expression pattern indicated that NEDD4 might function in mammalian testes. Recent reports have indicated a conserved function of NEDD4 (Rsp5) in target protein degradation under stress conditions[19]. As SG formation is known to protect germ cells from stress[24] and heat damage[25], we hypothesized that NEDD4 is involved in mouse spermatogenesis by regulating heat responses in germ cells. To test this hypothesis, we subjected both testes and cultured germline stem cells (GSCs) to heat stress at 42 °C for 20 min according to the previous report[25]. Given that the testicular temperature is maintained at 2–4 °C below the core body temperature[24], we used 33 °C as a control treatment (Supplementary Fig. 1D). By staining with the common SG marker TIAR and the germ cell-specific SG marker DAZL, we observed strong induction of SG formation[25] after 20 min of heat stress (Fig. 1b). Under the control temperature, NEDD4 was broadly observed in the cytoplasm, but NEDD4 was co-located with DAZL during heat stress (Fig. 1b). This co-localization with SGs occurred in both CDH1$^+$ and CDH1$^-$ cells (Supplementary Fig. 1E). The changes in subcellular localization, similar to those of SGs, were also observed for PABP1 (Supplementary Fig. 1F), which is a component of the mRNP complex and may be ubiquitinated by NEDD4 or Rsp5, as reported previously[19].

In addition, we found that NANOS2 was also colocalized with NEDD4 and DAZL (Fig. 1d). A NANOS2-IP assay showed that it interacted with NEDD4 and PABP1 in GSCs in its endogenous form (Fig. 1c and Supplementary Fig. 1G,H). Consistent with this result, a NEDD4 IP study also demonstrated that NEDD4 interacted with NANOS2, DAZL and PABP1, but not with TIAR, under both normal and heat-stress conditions (Fig. 1e). NEDD4 localization in speckles has been observed during its mediation of protein endocytosis through the endosomal–lysosomal pathway[26,27]. This recruitment is usually mediated by coactivators (NDFIP proteins)[28]. Staining with an early endosome marker, transferrin receptor (TfR) revealed that NEDD-4-positive SGs were localized to endosomes (Fig. 1f).

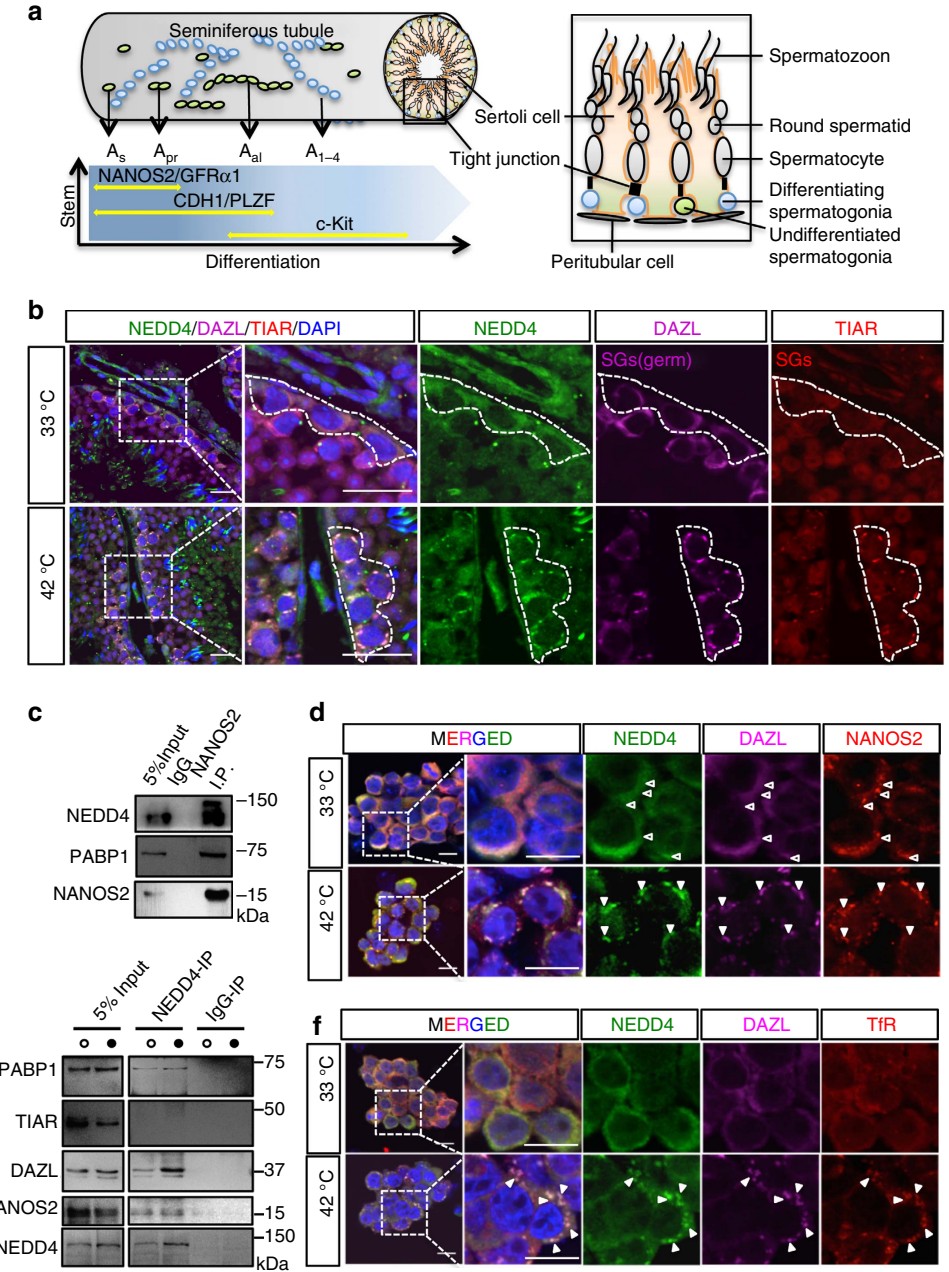

**Figure 1 | NEDD4 is expressed in male germ cells and heat stress changes subcellular localization of NEDD4.** (**a**) A schematic illustration showing the structure of a seminiferous tubule and the properties of early spermatogonial population depending on marker expression. The x axis denotes the differentiation stages. The y axis shows the self-renewal ability. (**b**) WT mice were incubated in 33 °C or 42 °C water baths for 20 min. Representative immunofluorescence (IF) images of testis slices stained for NEDD4, DAZL and the general SG marker TIAR are shown. The dotted line indicates spermatogonia, Scale bar, 20 μm; n = 3. (**c**) Western blots of a co-IP experiment with an anti-NANOS2 antibody. Cell lysates were prepared from GSCs. PABP1 was used as a positive control of the interaction with NANOS2 (ref. 3). Normal immunoglobulin G (IgG) served as a negative control, n = 3. (**d**) Cultured GSCs expressing FLAG-tagged NANOS2 were incubated at 33 and 42 °C for 20 min and then subjected to immunostaining for NEDD4 (green), FLAG (NANOS2, red) and the germ cell SG marker DAZL (magenta). Both NEDD4 and NANOS2 were colocalized with DAZL speckles after the heat stress. Open triangles indicate P bodies and filled triangles indicate SGs. Scale bar, 10 μm. (**e**) Western blots of a co-IP experiment with an anti-NEDD4 antibody and protein lysates from GSCs incubated at 33 °C (open circles) and 42 °C (filled circles) for 20 min. (**f**) GSCs subjected to the same treatment as in **d** were then stained for NEDD4, DAZL and TfR (an early endosome marker). Filled triangles indicate SGs. Scale bar, 10 μm.

These data suggest that germ cell mRNPs are dynamically modulated under normal and stress conditions, and that NEDD4 may be involved in this process.

**NEDD4 function in cultured SSC growth and differentiation.** To assess the possible role of NEDD4 in SSC maintenance, we conducted a knockdown experiment using cultured GSCs. We introduced *Nedd4* short hairpin RNAs (shRNAs) by means of the pSico lentivirus, a 4-hydroxytamoxifen (4-OHT)-inducible conditional knockdown (cKD) system, into *Rosa-CreER^T2* GSCs (Supplementary Fig. 2A,B). With this treatment, a reduction of Nedd4 at both the mRNA and protein levels was observed after 4 days (Fig. 2a–c). We then monitored the cell recovery rate

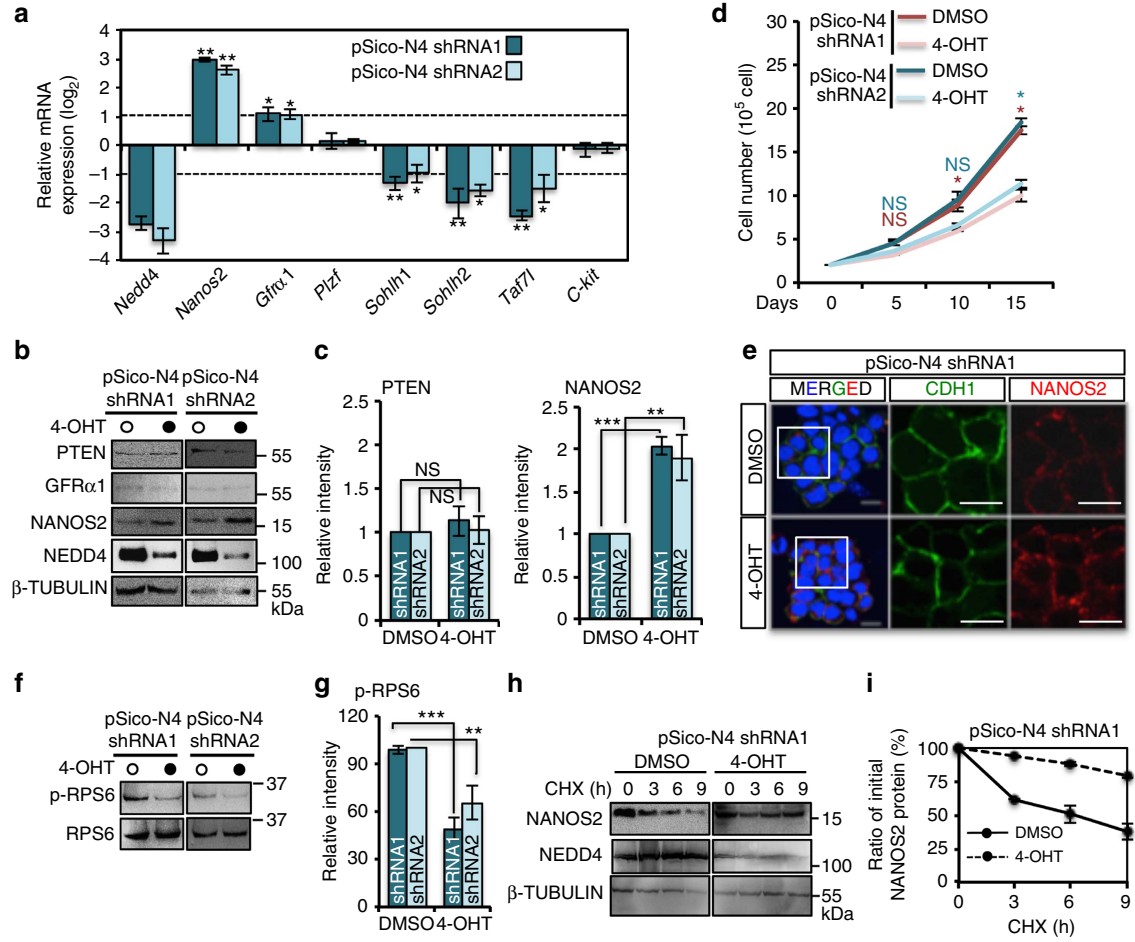

**Figure 2 | cKD of *Nedd4* prevents GSC differentiation.** (**a**) qPCR analysis of the key spermatogonial marker genes in pSico-*N4* shRNA1 or shRNA2 GSCs treated with 4-OHT for 96 h. For each gene, from the average mRNA level in GSCs treated with 4-OHT, we subtracted that in GSCs treated with DMSO ($n = 3$), and show them here as (log$_2$; ± s.d.). *$P < 0.05$; **$P < 0.01$, *t*-test. (**b**,**c**) pSico-*Nedd4* shRNA1 or shRNA2 GSCs treated with DMSO (vehicle) or 4-OHT (1 μM) were harvested for western blotting (**b**). The filled circles indicate 4-OHT treated groups. (**c**) Values of relative band intensity for PTEN and NANOS2 are shown as mean ± s.d. ***$P < 0.001$, **$P < 0.01$, *t*-test. (**d**) pSico- *Nedd4* shRNA1 or shRNA2 GSCs were treated with DMSO or 4-OHT and then passaged three times at 5-day intervals. The number of cells was counted to determine the cell recovery as an indicator of cell growth. The results are presented as mean ± s.d. ($n = 3$). The number of cells at each time point was compared by a *t*-test, *$P < 0.05$, *t*-test. (**e**) pSico- *Nedd4* shRNA1 GSCs treated with DMSO or 4-OHT for 96 h were fixed, and used for IF with anti-NANOS2 and anti-CDH1 antibodies. The number of NANOS2 foci was increased by the *Nedd4* knockdown. Scale bar, 10 μm, $n = 3$. (**f**,**g**) pSico- *Nedd4* shRNA1 GSCs treated with DMSO (vehicle) or 4-OHT (1 μM) for 96 h were harvested for western blotting to detect RPS6, phosphorylated RPS6. The data are presented as mean ± s.d. ($n = 3$). **$P < 0.01$, ***$P < 0.001$, *t*-test. (**h**,**i**) pSico-*Nedd4* shRNA1 GSCs were treated with DMSO (vehicle) or 4-OHT (1 μM) for 96 h and then incubated with CHX (20 nM) for the indicated periods. The cells were harvested for western blotting to detect the NANOS2 protein. (**i**) The intensity of bands was measured and shown as mean ± s.d. ($n = 3$).

of these GSC lines for 15 days (at 5-day intervals)[3,29] and found that *Nedd4*-cKD GSCs exhibited a lower rate of cell recovery (Fig. 2d and Supplementary Fig. 2C). We therefore checked gene expression using quantitative reverse transcription PCR and found that the expression of gene markers of spermatogonial self-renewal (including *Nanos2* and *Gfrα1*) was increased, while differentiation-related transcripts, such as *Sohlh1*, *Sohlh2* and *Taf7l*, were downregulated (Fig. 2a), suggesting that NEDD4 is required to promote SSC differentiation.

**Knockdown of *Nedd4* in SSCs increased NANOS2 protein levels.** Next, we identified possible targets of NEDD4 in SSCs. NEDD4 targets many proteins involved in cell proliferation and differentiation, such as PTEN, α-synuclein, and IRS1 (refs 26–28,30), for both proteasomal and lysosomal degradation. Among these targets, PTEN, a repressor of spermatogonial proliferation[31], did not accumulate in *Nedd4*-cKD GSCs (Fig. 2b,c), which is

consistent with reports showing that NEDD4 is dispensable for ubiquitination of PTEN (refs 32,33). NANOS2, a protein that interacts with NEDD4 (Fig. 1c), was upregulated in *Nedd4*-cKD GSCs (Fig. 2b,c). As NANOS2 suppresses both proliferation and differentiation of SSCs by regulating the fate of mRNAs[3,16], the decreased proliferation and differentiation in *Nedd4*-cKD GSCs may be partially attributed to the increased NANOS2 protein level. In line with this notion, the number of NANOS2 foci increased in *Nedd4*-cKD GSCs (Fig. 2e). Furthermore, ectopically expressed NEDD4 in GSCs reduced NANOS2 protein (Supplementary Fig. 2D) and increased expression of NANOS2 targets *Sohlh1*, *Sohlh2* and *Taf7l* (Supplementary Fig. 2E). In addition, further expression of NANOS2 repressed those differentiation targets (Supplementary Fig. 2D,E). These results indicate that NEDD4 is required for GSC differentiation by decreasing NANOS2 protein. The NANOS2-mRNP complex also inhibits the mTORC1 (mammalian target of rapamycin

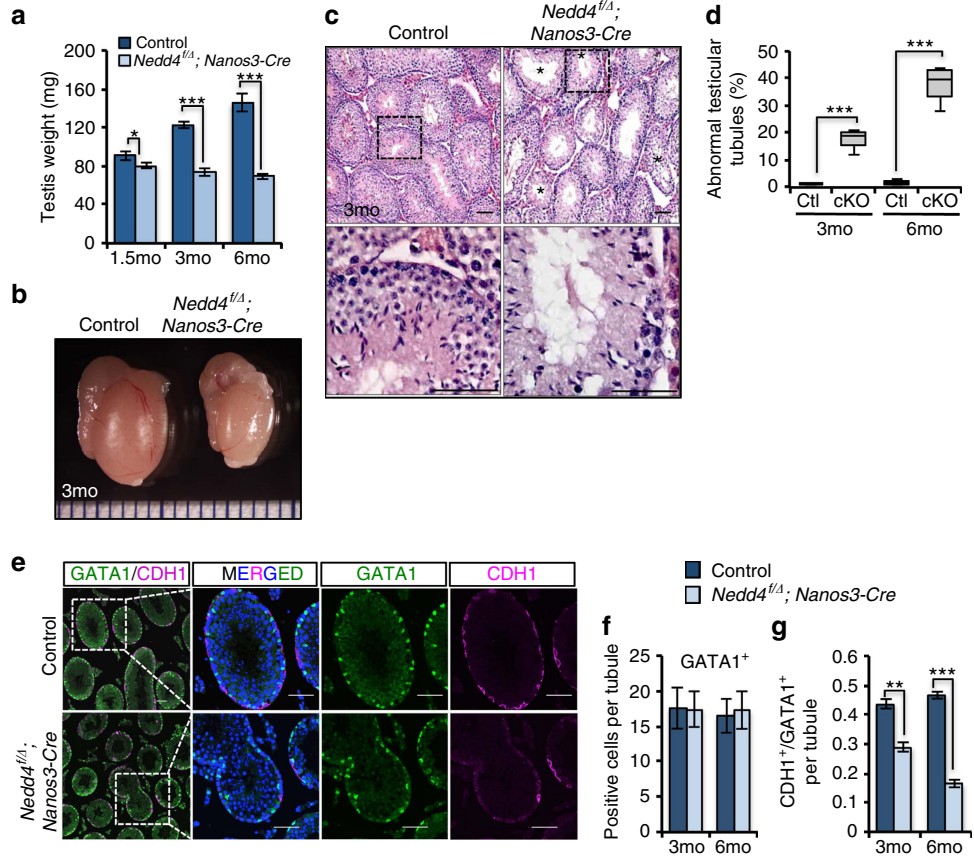

**Figure 3 | A loss of *Nedd4* results in an age-dependent defect of spermatogenesis.** (**a**) Testis weight was significantly reduced in *Nedd4* cKO mice in a time-dependent manner, $n \geq 3$ for each time point. *$P < 0.05$, ***$P < 0.001$, *t*-test. (**b**) A cKO of *Nedd4* in germ cells decreased testis size in 3-month-old mice. See also Supplementary Figs 3 and 4. (**c,d**) Testis slices from *Nedd4* cKO mice were examined after haematoxylin and eosin (H&E) staining. (**c**) Scale bar, 50 μm. (**d**) Abnormal testicular tubules with reduced germ cells (indicated by asterisks) from 3-month-old and 6-month-old mouse testes were counted and shown as mean ± s.d. ***$P < 0.001$, *t*-test. (**e–g**) Comparison of the number of GATA1$^+$ cells (**e,f**, Sertoli cells) and CDH1$^+$ cells (**e,g**, undifferentiated spermatogonia) per seminiferous tubule in control and *Nedd4* cKO mice. Scale bar, 50 μm. The data shown in **f,g** are the average of 40 tubules (± s.d.); **$P < 0.01$; ***$P < 0.001$, *t*-test.

complex 1) signalling, a pathway crucial for SSC proliferation and differentiation[3,34]. In *Nedd4*-cKD GSCs, phosphorylated RPS6, a major output for mTORC1 activity was reduced, (p-RPS6; Fig. 2f,g), which supported our previous observation. Given that *Nanos2* mRNA was upregulated in *Nedd4*-cKD GSCs, NANOS2 protein upregulation could be ascribed to the transcriptional regulation. To verify the translational regulation of NANOS2 via NEDD4, a cycloheximide (CHX) chase assay was performed. In the presence of NEDD4, the amount of NANOS2 protein decreased on CHX treatment, while in *Nedd4*-cKD GSCs, the amount of NANOS2 protein was stable (Fig. 2h,i), confirming that NEDD4 modulated NANOS2 protein stability.

**Nedd4 deletion reduces spermatogonial cells *in vivo*.** Whole-body deletion of *Nedd4* results in neonatal death[35], precluding analysis of *Nedd4*-deficient germ cells. Therefore to investigate the *in vivo* function of NEDD4 during spermatogenesis, we used a *Nedd4*-floxed allele[36] with the Cre recombinase under control of the *Nanos3* promoter (*Nedd4*$^{f/\Delta}$, *Nanos3-Cre*). CRE activity was germ cell specific and almost 100% at postnatal day 1 as shown by a floxed yellow fluorescent protein (YFP) reporter (Supplementary Fig. 3A). Immunohistochemistry demonstrated the deletion of NEDD4 protein in the *Nedd4* cKO mice (Supplementary Fig. 3B,C). Histological analyses of 6-week-old

*Nedd4*$^{f/\Delta}$;*Nanos3-Cre* mice showed no obvious defects, even though testes were smaller than those of controls (Fig. 3a). We next examined the time-dependent effects of conditional knockout (cKO) of *Nedd4* on testes. The testes from older *Nedd4* cKO mice (3 and 6 months of age) exhibited a marked reduction in weight (Fig. 3a,b), in line with the decreased size and cell proliferation observed in other organs[30,35]. Furthermore, histological results also showed an increased number of abnormal testicular tubules with a reduced fraction of germ cells in 3-month-old *Nedd4* cKO mice (Fig. 3c,d). These results suggest that *Nedd4*-null germ cells were detrimental for spermatogenesis, even though the *Nedd4* cKO mice were fertile at least until 6 months.

To clarify the function of NEDD4 in germ cells, we analysed germ cell properties in testes from 3- and 6-month-old *Nedd4* cKO mice. We confirmed that the GATA1$^+$ Sertoli cell numbers were unchanged in *Nedd4* cKO mice (Fig. 3e,f). Consistent with our observation that *Nedd4* was enriched in CDH1$^+$ SSC (Fig. 1), the CDH1$^+$ SPC number was reduced by deletion of *Nedd4* in male germ cells (Fig. 3e,g). The decrease in the SPC population was also revealed by PLZF staining (Supplementary Fig. 3D,E). PLZF is a factor important for control of the size of the SPC pool[11]. Consequently, the number of meiotic germ cells was also reduced in *Nedd4* cKO testes (Supplementary Fig. 3F). However, no obvious apoptotic germ cells were observed in *Nedd4* cKO

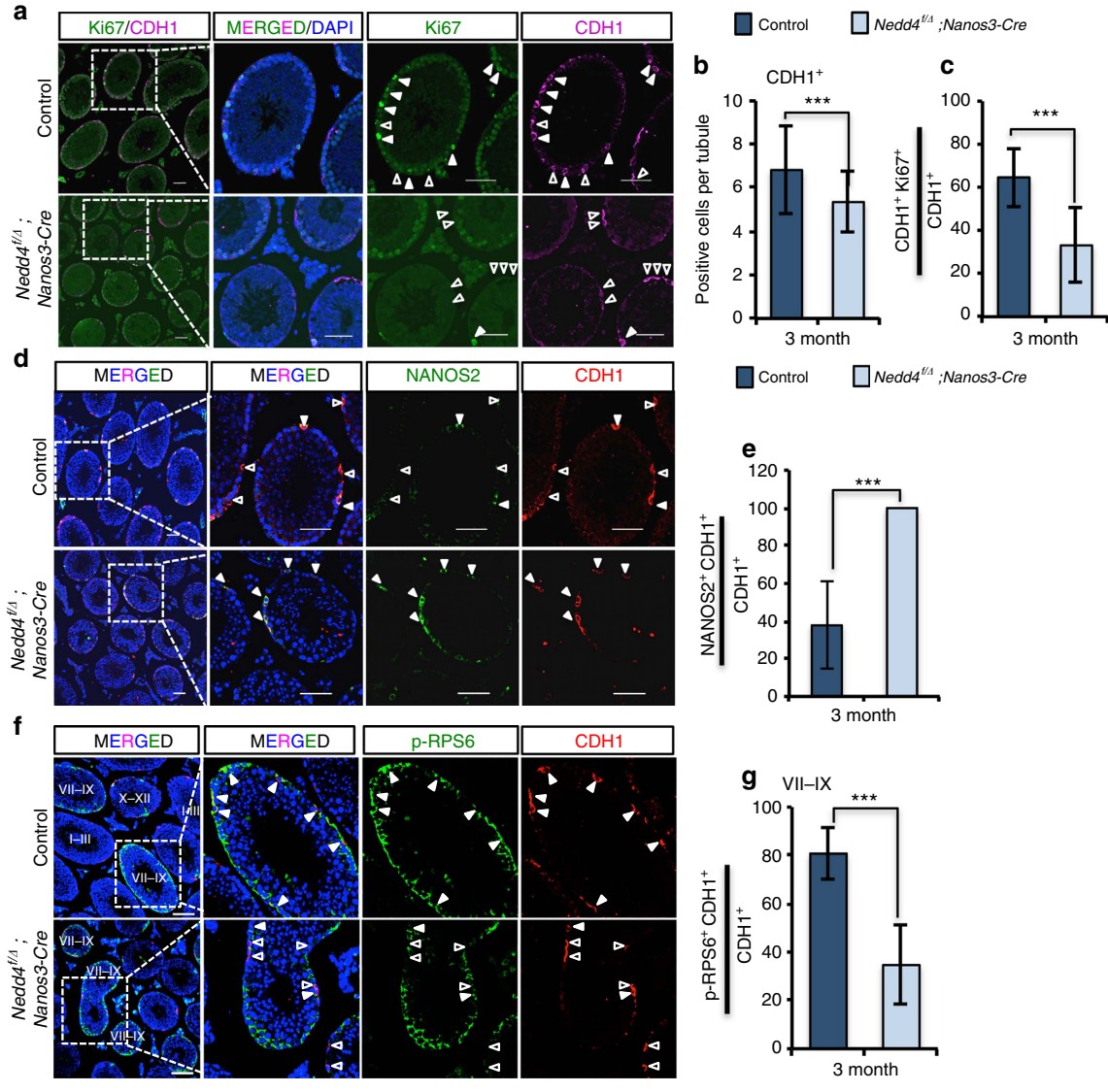

**Figure 4 | Loss of *Nedd4* increases NANOS2 stability and inhibits SPC proliferation.** (**a–c**) Representative pictures showing staining for CDH1 and Ki67 (a mitosis marker) of control and 3-month-old *Nedd4* cKO mice testis (**a**). Open triangles indicate CDH1$^+$Ki67$^-$ spermatogonia. Filled triangles indicate CDH1$^+$Ki67$^+$ spermatogonia. The average CDH1$^+$ cell number ($\pm$ s.d.) in 40 testicular tubules is shown in **b**. The proportion of proliferating SPCs (CDH1$^+$Ki67$^+$) among all SPCs was significantly decreased in testes from 3-month-old *Nedd4* cKO mice as shown in **c**; ***$P<0.001$, *t*-test. Scale bar, 50 μm. (**d,e**) Comparison of the number of NANOS2$^+$ cells per seminiferous tubule in control and *Nedd4* cKO mice. Open triangles indicate NANOS2$^-$CDH1$^+$ SSCs. Filled triangles indicate NANOS2$^+$CDH1$^+$ spermatogonia (most primitive SSCs). The proportion of NANOS2$^+$CDH1$^+$ cells among all CDH1$^+$ cells was significantly increased in *Nedd4* cKO testes as shown in **e**; ***$P<0.001$, *t*-test. Scale bar, 50 μm. (**f,g**) Phosphorylation of RPS6 is shown in both control and *Nedd4* cKO testes. Open triangles indicate p-RPS6$^-$CDH1$^+$ SSCs. Filled triangles indicate p-RPS6$^+$CDH1$^+$ spermatogonia. The proportion of p-RPS6$^+$CDH1$^+$ cells among all CDH1$^+$ cells was significantly decreased in *Nedd4* cKO testes as shown in **g**; testicular stages are indicated with Roman numerals. ***$P<0.001$, *t*-test. Scale bar, 50 μm.

testes (Supplementary Fig. 3G). These results suggest that deletion of *Nedd4* reduced the SPC pool size.

**NEDD4 regulates SSC homeostasis through NANOS2 *in vivo*.** We next investigated the cause of the reduction in the SPC pool. Given that NEDD4 is important for both proliferation and differentiation of cultured GSCs (Fig. 2), we compared the proliferation among those CDH1$^+$ progenitor cells by costaining for the mitosis marker Ki67. The number of CDH1$^+$ SPCs was reduced in *Nedd4* cKO testes (Figs 3g and 4a,b). The relative number of Ki67-positive cells in CDH1$^+$ SPCs was also reduced (by half) as compared with the number in

control mice (Fig. 4a,c). As NANOS2 is a strong inhibitor of both SSC differentiation and proliferation[3,16], and because NEDD4 was found to control the stability of NANOS2 in cultured GSCs (Fig. 2), we next examined NANOS2 expression *in vivo*. In control testes, NANOS2$^+$ SSCs were limited to the most primitive A$_s$ and A$_{pr}$ cells (Fig. 1a), which constitute ~30% of the CDH1$^+$ SPC pool. In contrast, this proportion was 100% in *Nedd4* cKO testes (Fig. 4d,e). Indeed, in *Nedd4* cKO testes, the NANOS2$^+$ population was greater than GFRα1$^+$ cells, even though they are almost the same population (most primitive A$_s$ and A$_{pr}$ SSCs) in control testes (Supplementary Fig. 4A). The NANOS2 protein was observed even in early meiotic cells in *Nedd4* cKO testes, but not in control testes

(Supplementary Fig. 4B). These results suggest that the NANOS2 protein was hardly cleared in the absence of *Nedd4*; thus, stabilized NANOS2 may be responsible for the reduction of SPC pool.

We also analysed mTORC1 signalling and found stage-dependent activation of p-RPS6 (Fig. 4f)[34] in both somatic and CDH1[+] spermatogonia in adult testes (Fig. 4f, Supplementary Fig. 4C,D) in agreement with the recent reports showing stronger activation of mTORC1 after RA treatment[34]. In contrast, mTORC1 signalling is suppressed in NANOS2[+] SSCs[3]. Therefore, we considered that clearance of NANOS2 was a prerequisite for RA-induced mTORC1 activation in CDH1[+] SPCs during the spermatogenic epithelial cycle. As expected, in the absence of *Nedd4*, mTORC1 activation was barely detectable in CDH1[+] SPCs, even in the RA-rich stages VII–XI of tubules (Fig. 4f,g). Therefore, the lower mitotic activity of SSCs must be due to the impaired mTORC1 activity by stabilized NANOS2 in *Nedd4* cKO mice, and this may be the cause of the smaller stem cell pool and the decreased testes mass. In conclusion, NEDD4 controls the SPC pool size by regulating NANOS2 stability during the normal epithelial spermatogenetic cycle.

**Nedd4 affects SG clearance after heat stress in GSCs.** Under heat-stress conditions, NANOS2 was recruited to SGs (Fig. 1d). As heat stress changed the subcellular localization of NEDD4 to SGs in GSCs, and NEDD4 interacted with mRNP components such as NANOS2, DAZL and PABP1, we hypothesized that NEDD4 modulates SG dynamics under heat stress. We first tested our supposition using *Nedd4*-cKD GSCs. After incubation of control and *Nedd4*-cKD GSCs at 42 °C, a number of SGs with NANOS2 was formed, and they were located in early endosomes (TfR; Supplementary Fig. 5A). When we returned the heat-shocked cells into a 33 °C incubator for 3 h (Fig. 5a), SGs with NANOS2 were disassembled in control GSCs, but this disassembly rarely occurred in *Nedd4*-cKD GSCs (Fig. 5b). Thus, we theorized that NEDD4 facilitates clearance of SGs after heat stress. To test this idea, we used two other SG markers, DAZL and TIAR. When GSCs were cultured at 33 and 37 °C, few SGs formed in either the control or *Nedd4*-cKD GSCs (Fig. 5c,f). SG formation was increased when those GSCs were cultured at 42 °C for 20 min (Fig. 5d,f). When these cells were returned to the 33 °C incubator, the majority of the SGs were cleared in the control GSCs, while 40–50% of SGs persisted in *Nedd4*-cKD GSCs even after 3 h of recovery (Fig. 5e,f). SG clearance defects are observed when the autophagy pathway is blocked or Valosin-containing protein (VCP), an ubiquitin segregase that helps extract polyubiquitinated proteins, is knocked down[2]. Thus, we next tested whether SG clearance is mediated by the endosomal–lysosomal pathway by treating GSCs with chloroquine (CLQ) after heat stress. CLQ treatment blocked almost all SG clearance (∼95%), similar to that seen in *Nedd4*-KD GSCs (Fig. 5g and Supplementary Fig. 5B,C). These results confirmed that the endosomal–lysosomal pathway mediates SG clearance in GSCs, and that NEDD4, an E3 ligase, is important for this process.

**Deletion of Nedd4 reduces thermotolerance of spermatogonia.** Based on the data from *Nedd4*-cKD GSCs, we next examined the function of NEDD4 in thermotolerance *in vivo*. Mammalian testes are located in the scrotum, which has a temperature below the core body temperature. Artificially induced crypto-rchidism or forced heat stress on the testes causes apoptosis of germ cells[25,37]. On the other hand, the spermatogonia population under heat-stress conditions induces SG formation to resist

heat stress[25]. We next studied whether both rapid formation of SGs during stress and dynamic clearance of SGs soon after stress would contribute to the survival of SSCs. We first examined the effects of long-term heat stress on spermatogenesis (Supplementary Fig. 6A). A single heat stress (20 min) did not affect spermatogenesis, but repeated heat stress (7–21 days, 20 min per day) reduced testes weight and spermatogenesis (Supplementary Fig. 6B,C). We then compared apoptosis among different germ cell populations. Increased c-PARP staining was observed in SYCP3[+] meiotic cells after 7 days of heat stress (Supplementary Fig. 6D,E), while most CDH1[+] cells did not undergo apoptosis (Supplementary Fig. 6F,G). This result is consistent with a previous study showing that spermatogonia are more resistant to heat stress because of rapid formation of SGs[25]. To test whether NEDD4 is also involved in the clearance of SGs upon heat stress *in vivo* as seen in GSCs, we decided to compare the responses in the control and *Nedd4* cKO testes after 7 days of heat stress (Fig. 6a). After a 1-day recovery, SGs were disassembled successfully in control testes but not in *Nedd4* cKO testes (Fig. 6b). In addition, a greater proportion of CDH1[+] SPCs underwent apoptosis in *Nedd4* cKO testes than in control testes (Fig. 6c,d). This phenomenon resulted in a further reduction of the CDH1[+] SPCs pool by *Nedd4* deletion (Fig. 6c,e). Similar results were obtained with cultured GSCs. Under normal conditions, a low level of NEDD4 did not affect survival of GSCs. During heat stress, however, *Nedd4*-cKD GSCs formed SGs normally, but they had difficulty in clearing the SGs and started to die. As a result, during the 4-hr recovery period, the apoptosis index increased in *Nedd4*-cKD cells as compared with control cells (Fig. 6f,g). To further trace the defects of the CDH1[+] SPC pool in *Nedd4* cKO testes, we used a 30-day recovery time, which is almost equal to one round of spermatogenesis[15]. In control testes, CDH1[+] cells were protected from heat stress and subsequently recovered, thus spermatogenesis was well reconstituted after 1 month (Supplementary Fig. 6H,I). In contrast, damaged CDH1[+] stem cells by heat stress in *Nedd4* cKO testes resulted in marked loss of germ cell development (Supplementary Fig. 6H), and the abnormal testicular tubules with reduced germ cell reached to ∼70%, whereas it was only 30% in the littermates of *Nedd4* cKO mice without heat treatments (Supplementary Fig. 6I). Our results demonstrate that NEDD4 safeguards SSCs from heat stress to secure long-lasting spermatogenesis in males.

**NEDD4 directly targets NANOS2 for degradation.** Our experiments indicate that NEDD4 is required for the main-tenance of the SPC pool in normal spermatogenesis, and it is important to clear SGs formed in the heat-stress conditions. To explore the underlying molecular mechanism of how NEDD4 works, we investigated the relationship between NEDD4 and NANOS2 in more detail. We expressed MYC-NEDD4 and FLAG-NANOS2 in HEK 293T cells and examined the interac-tion. The results showed that NEDD4 was coimmunoprecipitated with NANOS2 (Fig. 7a), as was expected, based on the endogenous interaction of NEDD4-NANOS2 in GSCs (Fig. 1c,e). Subsequently, we performed an *in vitro* pull-down assay using glutathione S-transferase (GST)-NEDD4 and MBP-NANOS2 purified from *Escherichia coli,* and confirmed the direct interaction of NEDD4 and NANOS2 (Fig. 7b). To further confirm the interaction *in vivo* and to visualize the dynamics of protein interaction, we conducted a bimolecular fluorescence complementation (BiFC) assay[38] (Fig. 7c). After transfection of the plasmids encoding N-terminal and C-terminal nonfluorescent VENUS (NTV and CTV) fragments to NEDD4 and NANOS2, respectively, we monitored the signals from reconstituted VENUS

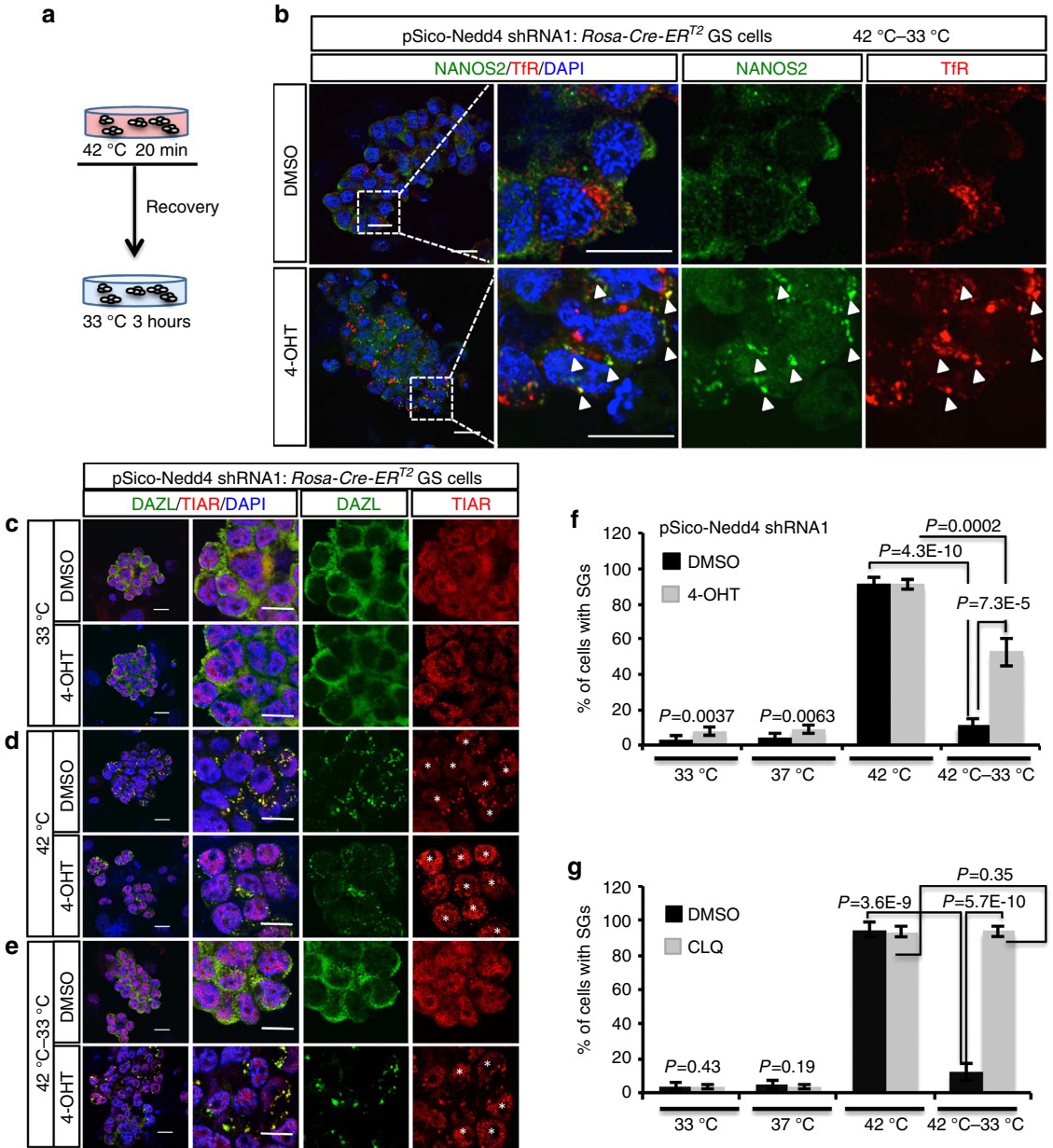

**Figure 5 | NEDD4 is required for SG clearance after heat stress. (a)** A schematic showing the heat stress (42 °C, 20 min) and recovery treatment (33 °C for 3 h) of GSCs. **(b)** pSico-*Nedd4* shRNA1 GSCs treated with DMSO or 4-OHT for 96 h were treated with heat stress and then allowed to recover as shown in **a**. GSCs were immunostained with anti-NANOS2 and anti-TfR antibodies ($n = 3$). Arrowheads indicate merged foci. Scale bar, 10 μm. **(c–g)** pSico- *Nedd4* shRNA1 GSCs treated with DMSO or 4-OHT were incubated at 33 °C for 20 min **(c)**, 42 °C for 20 min **(d)** or allowed to recover (E, 42–33 °C). SGs were detected by staining for DAZL and TIAR. *GSCs with SGs. Scale bar, 10 μm. **(f)** Quantitation of the data from **(c–e)**. The results are presented as mean ± s.d. ($n = 5$), *t*-test. **(g)** Culture conditions were similar to those in **c–e**, and incubation with CLQ (lysosome inhibitor) was performed 2 h before the heat stress. Statistical analysis of GSCs treated with DMSO (control) or CLQ is shown. The results are presented as mean ± s.d. ($n = 5$), *t*-test.

as readout of NANOS2 and NEDD4 interaction (Fig. 7c–g and Supplementary Fig. 7A,B). As NANOS2 is localized to PBs in male germ cells[39], we used the PB marker RCK to determine whether the NANOS2-NEDD4 complex was localized to PBs. We found that some portion of NANOS2 localized to RCK-positive P-bodies. However, the signal from the VENUS protein did not co-localize with the RCK signals, indicating that the interaction between NANOS2 and NEDD4 occurred in other organelles (Fig. 7e, Supplementary Fig. 7B). Accordingly, we tested other markers and found that VENUS signals colocalized with endosome (TfR, Fig. 7f and Supplementary Fig. 7B) and lysosome

(LAMP1, Fig. 7g and Supplementary Fig. 7B) markers, indicating that the NANOS2-NEDD4 complex was recruited to the endosome-lysosome pathway, which supports our previous results using GSCs (Fig. 1f, Supplementary Figs 4 and 5A).

Next, we investigated the possibility of NANOS2 being a direct target of NEDD4 and that NANOS2 could be ubiquitinated by NEDD4. To test this possibility, we conducted an *in vitro* ubiquitination assay. MBP-NANOS2 was incubated with or without GST-NEDD4, or ubiquitin, and we found that signals of high molecular weight appeared in western blotting using the anti-NANOS2 antibody only when MBP-NANOS2 was incubated

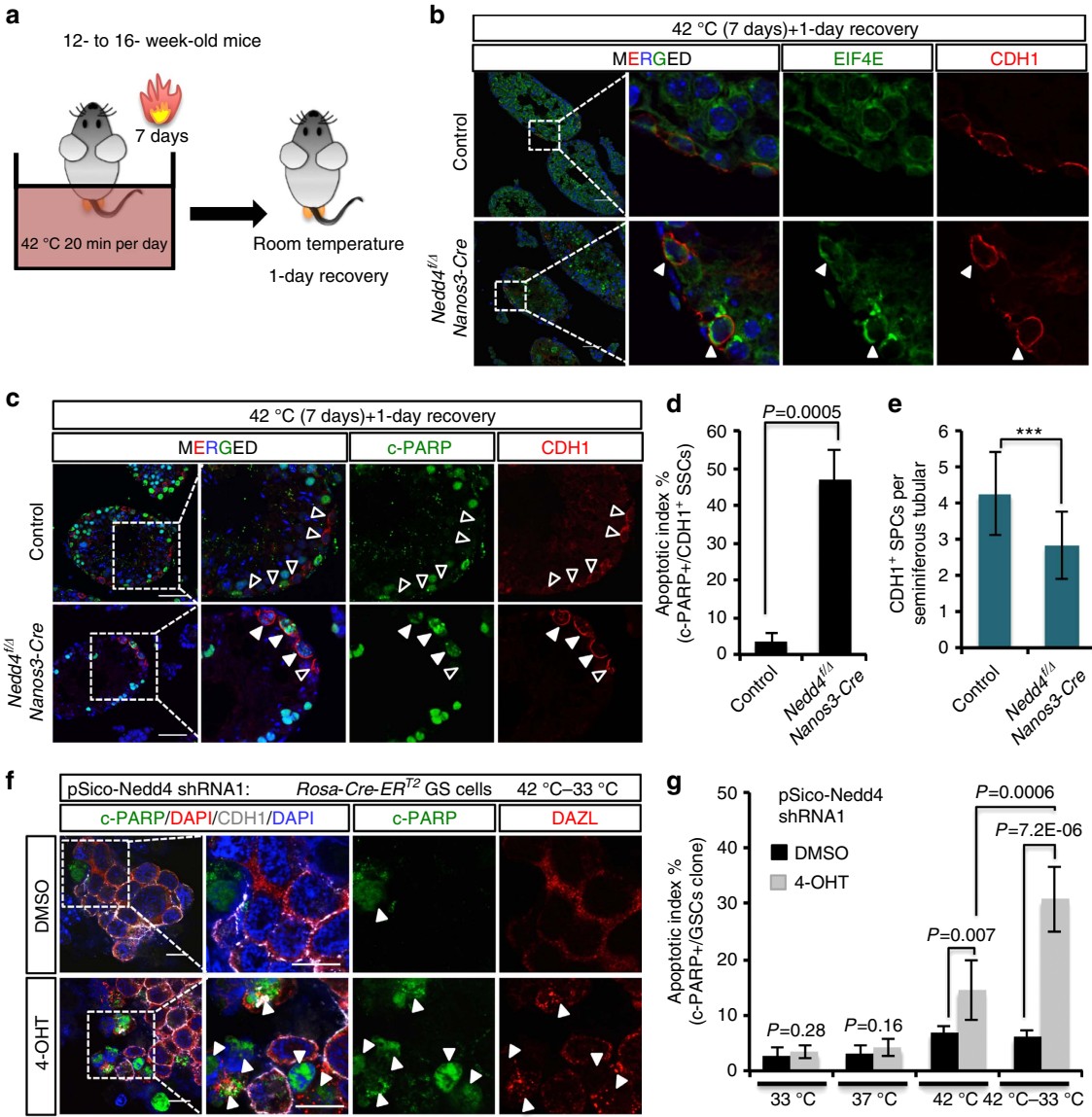

**Figure 6 | NEDD4 is required for thermotolerance of mouse testes.** (**a**) A schematic illustration showing long-term heat stress *in vivo*. WT or *Nedd4* cKO mice at 12–16 weeks of age were anaesthetized and incubated in a 42 °C water bath for 20 min per day. After 7 days of this treatment, the mice were allowed to recover for 1 day before euthanasia. (**b**) Representative pictures showing the staining for EIF4E (SG marker) and CDH1 (SPC marker) after the treatment described in **a**. A large number of stable SGs persisted only in *Nedd4* cKO testes. Arrowheads indicate SPCs with clear-cut staining of SG foci. Scale bar, 50 μm. (**c–e**) Increased apoptosis was observed in *Nedd4* cKO SPCs. Costaining for CDH1 and c-PARP was performed and representative pictures are shown (**c**). Open arrowheads indicate c-PARP⁻CDH1⁺ SPCs; filled arrowheads indicate c-PARP⁺CDH1⁺ SPCs. Statistical analysis is shown for mean ± s.d. in **d**. CDH1⁺ SPCs were counted and shown as mean ± s.d. in **e**; ***$P < 0.001$, $t$-test. Scale bar, 50 μm. (**f,g**) Representative pictures showing a greater extent of c-PARP staining (apoptosis) during the 42–33 °C recovery (4 h) in *Nedd4*-cKD GSCs than in the control group. Arrowheads indicate apoptotic cells. Statistical analysis is shown for mean ± s.d. in **g** ($n = 3$), $t$-test. Scale bar, 10 μm.

with GST-NEDD4 and ubiquitin (Fig. 7h). Interestingly, the level of the parent band of MBP-NANOS2 showed a clear reduction after the ubiquitination reaction. To further test whether NANOS2 could be ubiquitinated by endogenous NEDD4 or not, we introduced MYC-NANOS2 to mouse embryonic fibroblast (MEF) cells established from both wild type (WT) and *Nedd4* knockout (KO) mice. NANOS2 was successfully ubiquitinated in WT MEF cells, but this ubiquitination was abrogated in *Nedd4* knockout MEF cells (Supplementary Fig. 7C). These results indicate that NEDD4 has the potential to ubiquitinate NANOS2 efficiently.

Given that NEDD4 recruitment is known to be mediated by the coactivator of NEDD4, NDFIP2 (ref. 28), we investigated

localization of NDFIP2 using the BiFC assay and found that NDFIP2 also colocalized with the NEDD4/NANOS2 complex (Supplementary Fig. 7A,B), further supporting our hypothesis that the NANOS2 protein is degraded via a NEDD4/NDFIP2-mediated endosomal–lysosomal pathway. To examine whether complex formation of NEDD4 and NDFIP2 promotes NANOS2 ubiquitination *in vivo*, FLAG-tagged NANOS2 was expressed in HEK293 cells with or without MYC-NEDD4 and MYC-NDFIP2. Immunoprecipitated FLAG-NANOS2 was subjected to SDS–polyacrylamide gel electrophoresis (SDS–PAGE) and western blotting using anti-FLAG, anti-MYC and anti-ubiquitin antibodies (Fig. 7i). The protein complex including FLAG-NANOS2 was ubiquitinated by MYC-NEDD4,

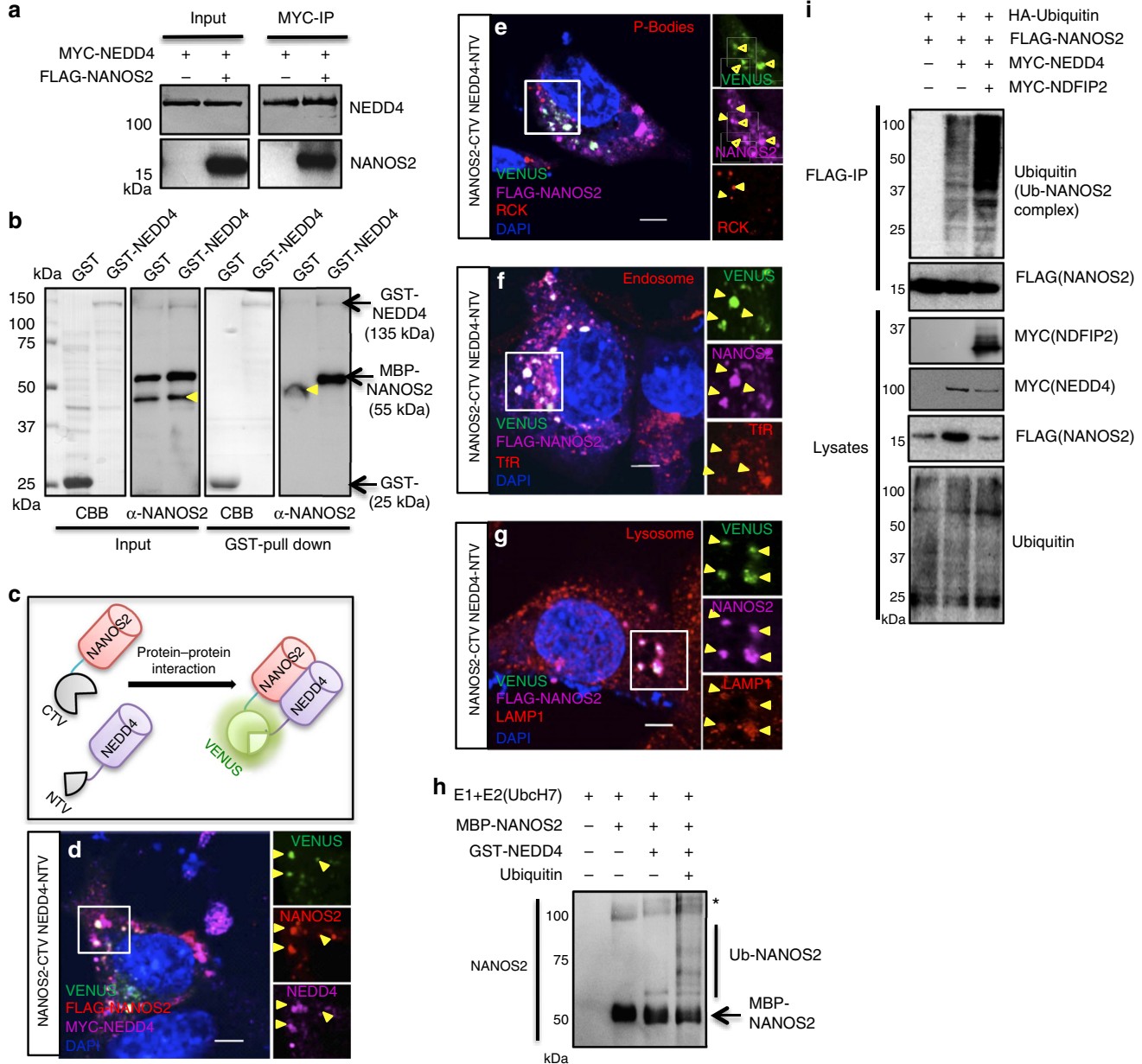

**Figure 7 | NEDD4 directly targets NANOS2 for degradation. (a)** Western blot analysis of NANOS2 immunoprecipitated by an anti-MYC (NEDD4) antibody in HEK 293T cells. **(b)** Western blot analysis of MBP-NANOS2 purified with either GST or GST-NEDD4. Yellow arrowheads indicate nonspecific bands ($n = 3$). **(c)** A diagram of the method used for the *in situ* analysis of the interaction between NEDD4 and NANOS2 by means of a bimolecular fluorescent complementation assay (BiFC). CTV: C terminus of VENUS protein, NTV: N-terminus of VENUS protein. **(d–g)** Analysis of NIH3T3 cells transfected with FLAG-NANOS2-CTV and MYC-NEDD4-NTV. Interaction between NANOS2 and NEDD4 was visualized by VENUS signals. The NANOS2 and NEDD4 proteins were immunostained as red and magenta, respectively (**d**). Dcp1a (P body marker) staining was colocalized with NANOS2-only foci but not NANOS2-NEDD4 double-positive foci (**e**). Open triangles indicate NANOS2 single-positive foci. Filled triangles indicate NANOS2-NEDD4 double-positive foci, which are not colocalized with Dcp1a. Scale bar, 5 μm. (**f,g**) The interaction of NEDD4 and NANOS2 (VENUS) was observed in both endosomes (**f**, TfR) and lysosomes (**g**, LAMP1). Three independent transfections were performed. Scale bar, 5 μm. (**h**) Western blot analysis showing that NEDD4 directly promotes MBP-NANOS2 ubiquitination *in vitro*. (**i**) Western blot analysis of 293T cells cotransfected with plasmids expressing HA-tagged ubiquitin, FLAG-tagged NANOS2, and MYC-tagged NDFIP2 and NEDD4 ($n = 3$). IP was performed with an anti-FLAG (NANOS2) antibody.

and overexpression of MYC-NDFIP2 further promoted this ubiquitination (Fig. 7i). Furthermore, we found that a GSC differentiation-inducing factor, RA, induced expression of NDFIP2, which may enhance the degradation of NANOS2 protein (Supplementary Fig. 7D). These results are consistent with previous observations that NDFIP2 acts as a coactivator to promote auto-ubiquitination of the NEDD4/NDFIP2 complex and degradation of its targets through the lysosomal

pathway[28,40]. Our previous data demonstrated that a mutant NANOS2 lacking N-terminal amino acids (ΔN10) has a much higher stability in Hela cells[41], indicating that the N-terminus of NANOS2 functions as a degron. By Co-IP assay, we found that ΔN10-NANOS2 rarely interacted with both NEDD4 and NDFIP2 (Supplementary Fig. 7E,F). We further confirmed this by performing CHX chase assay and found that ΔN10-NANOS2 was much more stable than WT NANOS2

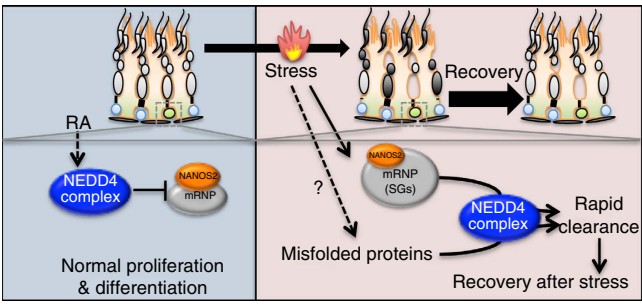

**Figure 8 | NEDD4 function in adult spermatogonia.** Illustration showing regulation pathways of the NANOS2-mRNP mediated by the NEDD4 complex in mouse spermatogonia. Under normal (left) condition, NEDD4 complex promotes NANOS2 degradation and spermatogonia differentiation. Under heat stress (right) conditions, NEDD4 complex mediates rapid clearance of SGs or misfolded proteins, thereby protecting spermatogonia from heat stress.

(Supplementary Fig. 7G,H). Taken all together, we concluded that the N-terminal of NANOS2 serves as a binding region for the NEDD4/NANOS2 complex, thereby leading it to lysosomal degradation.

## Discussion

An elaborate post-transcriptional network involving mRNP granules controls the fate of many adult stem cells[1,6]. We recently suggested that NANOS2-mRNP complexes establish a 'post-transcriptional buffer' system in SSCs to maintain their stemness[3,16]. However, the mechanisms underlying the dynamic regulation of mRNP granules are poorly understood. In the present study, we demonstrated that the E3 ubiquitin ligase, NEDD4, modulates mRNP dynamics in SSCs under both normal and heat-stress conditions (Fig. 8). In the absence of NEDD4, mRNP proteins like NANOS2 were stabilized in differentiating SSCs, thereby slowing down the general proliferation and differentiation of SSCs. Under heat-stress conditions, the dynamics of mRNPs facilitate the stress response in SSCs. The rapid formation of mRNP SGs is known to be important for germ cell survival during heat stress[25]. Our findings indicate that clearance of SGs is also necessary for the cells to recover from heat stress. This process is driven by NEDD4-mediated endosomal–lysosomal degradation.

We found that one important function of NEDD4 in SSCs is regulation of the state of mRNPs by targeting NANOS2 and/or other substrates for degradation. Given the strong inhibitory effects of NANOS2 on SSC differentiation, clearance of NANOS2 should be a prerequisite for progression to differentiation[16]. We believe that the negative regulation of NANOS2 by NEDD4 plays a key role in the initiation of SPC differentiation. During spermatogenesis, RA signalling acts as an inducer of differentiation[42]. We showed that NDFIP2, a target of NEDD4 was upregulated by RA signalling. NDFIP2 also recruits NEDD4 family proteins to endosomes or lysosomes and functions as a coactivator to promote degradation of target proteins[28]. Thus, we propose that the RA-NEDD4 complex cascade enhances NANOS2 protein degradation, thereby initiating SSC differentiation. NANOS2 effectively captures and silences target mRNAs, such as *Sohlh1*, *Sohlh2* and *Taf7l,* which are important for spermatogonial differentiation[3,39]. In the absence of *Nedd4*, however, those mRNAs are still captured by the stabilized NANOS2 protein, and this situation inhibits differentiation. Consequently, the NANOS2 targets were hardly translated. Therefore, spermatogonia were arrested in a NANOS2-positive state with poor differentiation.

Mammalian SGs are known to be cytoplasmic, nonmembranous aggregates that are assembled in response to stressors such as heat or toxic chemicals[1,25,43]. In the present study, we provide lines of evidence to show that dynamic regulation of SGs by the *Nedd4* complex plays a crucial role in thermotolerance of SSCs. First, we found that, after heat stress, most SGs were colocalized with NEDD4 and with endosomes or lysosomes (Fig. 1), which are typical membrane structures usually involved in endocytosis and protein degradation. Second, both *Nedd4*-cKD GSCs and *Nedd4* cKO testes exhibited a decreased ability to clear SGs after heat stress (Figs 3 and 6). Third, we identified several SG components, such as NANOS2, DAZL and PABP1, which are modulated by NEDD4; among them, NANOS2 was directly targeted by NEDD4 for degradation. Finally, in the absence of NEDD4, SSC survival was reduced.

We demonstrated that NEDD4 is involved in the regulation of major components of germ cell mRNPs. This process must be important for targeting SGs to the autophagy pathway[2] because heavy ubiquitination of SGs is observed in mammalian cells[44] and knockdown of VCP, an ATPase for extraction of ubiquitinated proteins from the cellular complex for degradation, blocks the clearance of SGs[2,45,46]. In SSCs, strong enrichment of NEDD4 may mediate ubiquitination of SG components in germ cells; thus, after stress, specific components must be selectively guided for lysosomal degradation. In addition, NEDD4 is the major E3 ligase for protein quality control after heat stress. The most important mediator of the interaction of NEDD4 and its targets in this process is the adaptor complex Hsp40-Ydj1 (ref. 19). A recent study indicated that the Hsp40-Ydj1 complex colocalizes with SGs after heat stress and is necessary for the clearance of these granules[47]. Furthermore, a loss of function of *DjA1*, the mouse homologue of *Ydj1*, leads to a severe defect in spermatogenesis[48]. Our study and others have shown that SSCs exhibit greater heat-stress resistance than differentiated cells[25]. Based on the above observations, we believe that this phenomenon can be attributed to greater NEDD4 activity, by which mRNPs are recruited to SGs after heat stress and are rapidly cleared in the recovery period. The regulation of mRNPs by the NEDD4 complex ensures sufficient flexibility in the stress response of SSCs.

In conclusion, our data provide evidence of a protein degradation pathway mediated by NEDD4 that regulates mRNP complexes, and controls adult stem cell differentiation and survival. This mechanism may provide insights into other areas of adult stem cell research and/or pathogenesis of stress-induced diseases.

## Methods

**Mouse maintenance and manipulations.** *Rosa-CreER^T2* and *Nanos3-Cre* mice were maintained and used as a tool to conditionally induce gene deletion[21]. *Nedd4^flox/flox* mice were used to delete *Nedd4* specifically in germ cells[36]. All mice were maintained in a C57BL/6/MCH background; we used mice less than 1 year old for phenotype analysis. For heat-stress experiments, 12–16 weeks old male mice were used. To administer heat stress to the mice, we followed a reported protocol[25]. The mice were anaesthetized and then placed in a 33 °C or 42 °C water bath; each heat stress lasted for 20 min per day (and some mice received daily heat stresses for 7–21 days). All animal experiments were conducted with the approval of the Institutional Animal Care and Use Committee of the National Institute of Genetics, Japan.

**Cell culture.** MEFs derived from C57BL6 embryos (isolated at day 13.5 post-coitum) were used as feeders for GSC culture. MEF cells treated with mitomycin-C were plated on 0.1% gelatin-coated dishes before GSC culture. Ros*a-CreER^T2* GS cell line was generated in our lab. To establish Ros*a-CreER^T2* GSC lines, postnatal mouse testes (day P5) from *Rosa-CreER^T2* mice were digested into cell suspension and then seed into 12-well plates for colony formation. GSC colony were moved to feeder cells and cultured with StemPro medium (Thermo) supplement with rat glial cell line-derived neurotrophic factor (20 ng ml$^{-1}$, R&D systems), mouse EGF (20 ng ml$^-$, BD Bioscience) and human FGF2 (10 ng ml$^-$, Life Technology). To induce CRE activity in GSCs carrying *Rosa-CreER^T2*, 4-OHT (1 µM) was added to the GSC medium at the indicated time

points. RA (100 nM; Wako) was used to induce mTORC1 signalling and GSC differentiation. CLQ (10 μM; Sigma-Aldrich) was used to inhibit lysosomal degradation. The 293T cell line (CRL-3216) purchased from ATCC was used for lentivirus production.

**Plasmids.** Full-length sequences of *Nanos2*, *Nedd4* and *Ndfip2* were subcloned from a GSC cDNA library into the pCMV-FLAG and pcDNA3.1-MYC plasmids. *Nanos2* and *Nedd4* were subcloned in to pCSII-EF for lentivirus production. For the GST pull down assay, *Nedd4* was subcloned into the pGEX-5X vector (GE Healthcare). Constructs were verified by sequencing.

**Immunofluorescence and immunohistochemistry.** Cultured GSCs grown on MEF cells in chamber slides (8 well, IWAKI) were washed in PBS, fixed in 4% paraformaldehyde for 15 min at room temperature then permeabilized in PBS with 0.3% Triton-X 100 for 15 min. After washing with PBST (PBS with 0.1% Tween 20), cells were blocked using 5% BSA in PBS before incubation with primary antibodies overnight at 4 °C. Testes for IF were fixed in 4% paraformaldehyde overnight at 4 °C and then embedded in paraffin or optimum cutting temperature compound (O.C.T.), and processed for haematoxylin and eosin and immunostaining. After primary and secondary antibody incubation, slides were counterstained with DAPI and mounted in the mountant permaFluo (Thermo), then analysed using an Olympus FV1200 confocal microscope (Olympus). Antibodies used are described in the Supplementary Methods.

**Coimmunoprecipitation and western blotting.** For the mass spectrometry analyses, 100 testes from either WT or transgenic E15.5 embryos expressing FLAG-tagged NANOS2 were used for generation of extracts. The supernatants were then mixed with 30 μl of anti-FLAG M2 affinity beads (Sigma-Aldrich) and incubated on a rotator for 3 h at 4 °C. After 5 washes with wash buffer, the beads were boiled in sample buffer, and then, the eluates were separated by SDS–PAGE and visualized with the Silver Quest silver staining kit (Invitrogen). Both gels for WT and transgenic lanes at the same molecular weight were excised, and then analysed by the mass spectrometry unit of the Center for Developmental Biology, RIKEN. The resulting tryptic peptides of mass data were matched against the NCBInr database using the Mascot or Sequest programs. The proteins only in the transgenic gel but not in the WT gel were identified as candidates for NANOS2-associated proteins.

For the IP from HEK 293T cell extracts, MYC-tagged NEDD4 or Ndfip2, FLAG-tagged NANOS2, and NANOS2 mutant and HA-tagged ubiquitin were cloned into pcDNA3.1 (Invitrogen), and then transfected into HEK 293T cells. The supernatants were then mixed with 20 μl of anti-FLAG M2 affinity beads (Sigma) and incubated on a rotator overnight at 4 °C. After five washes with wash buffer as previously described, the beads were boiled in sample buffer and the eluates were analysed by western blotting analyses. For the IP from WT and Nedd4 KO MEF cell extracts, MYC-NANOS2 was transfected. Cells were then lysate with SDS containing lysis buffer; the supernatants were mixed with anti-MYC antibody and incubated with on a rotator overnight at 4 °C. After 5 washes, the beads were boiled in sample buffer and the eluates were analysed by western blotting analyses. GSCs (~1 × 10^6 cells) were gently collected, washed in ice cold PBS before lysing in an equal volume of lysis buffer containing 50 mM Tris pH 7.4, 1% NP-40, 0.25% Na-deoxycholate, 150 mM NaCl and 1 mM EDTA supplemented with Protease inhibitor cocktail (Roche) and 0.1 mM PMSF (Wako), for 1 h on ice, and centrifuged at 20,000g for 10 min. Supernatant was mixed with 2 × loading buffer and applied to an SDS page gel. Antibodies used are described in the Supplementary Information. Horseradish peroxidase conjugated secondary antibodies were from Cell Signaling Technology. Quantification of scanned blots was performed using ImageJ software. Uncropped scans of the most important western blots were shown in Supplementary Fig. 8.

**Nedd4 shRNA knockdown.** Nedd4 shRNAs were designed by online software from Invitrogen as previously reported[3]. After screening, effective Nedd4 shRNA sequences were inserted in the Lentiviral vector (pSICO; Addgene). Insert oligos were as follows: Nedd4 shRNA1: 5′- T GGGCTTGTGTAATGAAGATCATTCA-AGAGATGATCTTCATTACACAAGCCCTTTTTTC -3′; 5′-TCGAGAAAAA-AGGGCTTGTGTAATGAAGATCATCTCTTGAATGATCTTCATTACACA-AGCCCA -3′. Nedd4-shRNA2: 5′- T GCACATCCTTCTGAAACTACTTTCAA-GAGAAGTTTCAGAAGGATGTGCTTTTTTC -3′; 5′-TCGAGAAAAAG-CATCCTTCTGAAACTACTTCTCTTGAAAGTTTCAGAAGGATGTGC A -3′. A pSICO vector containing scramble shRNA was used as negative control. For generation of the lentivirus, 293T cells were transfected with a combination of pSICO vector and packaging plasmids using PEI (Sigma) according to the manufacturer's instructions. Lentivirus-containing supernatant was collected in GSC medium 48 h after transfection. GSCs (*Rosa-CreER^{T2}*) were infected with the lentivirus supplemented with 6 μg ml^{-1} polybrene (Sigma). Four days after infection, cells were sorted by green fluorescent protein and re-plated to establish stable GSC lines. After expansion, cells were used to conditionally induce Nedd4 or scramble shRNAs.

**Quantitative reverse transcription polymerase chain reaction.** Total RNA was harvested from cells and tissue using Trizol reagent (Invitrogen) according to the

manufacturer's instructions. Cultured GSCs were gently washed off from feeder cells before resuspending in Trizol. RNA samples were further treated with DNAase I (Ambion) and then used for first strand cDNA synthesis using a Superscript III first strand synthesis kit (Invitrogen). For quantitative PCR reactions on cDNAs, a KAPA SYBR FAST qPCR Kit was used together with gene-specific primers. Primers used are described in the Supplementary Table 1.

**GST pull down assay.** MBP-NANOS2 protein was expressed in the E.coli BL21 (DE3) strain and purified with Amylose Resin (New England Biolabs). GST-NEDD4 and GST proteins were expressed in the E.coli BL21 Star (DE3) strain. Bacterial pellets were sonicated in a binding buffer (25 mM HEPES-KOH [pH 7.4], 150 mM NaCl, 0.1% NP-40, 1 mM DTT, 1 mM EDTA and 1 mM PMSF). The supernatants were mixed with 1 mg of MBP-NANOS2 for 2 h at 4 °C and then mixed with glutathione-sepharose 4FF (GE Healthcare) followed by further incubation for 2 h. The precipitates were separated by SDS–PAGE and analysed by western blotting with anti-NANOS2 antibody or by CBB (Coomassie brilliant blue) staining.

**Ubiquitination assay.** The *in vitro* ubiquitination assay was conducted in a total volume of 25 μl consisting of 50 mM Tris, 0.5 mM ATP, 100 nM E1 (Enzo), 1 uM E2 UbcH7 (Enzo), 2.5 μM ubiquitin, 0.5 μg GST-NEDD4 and 2 μg MBP-NANOS2. The reaction mixture was incubated at 30 °C for 3 h. The reaction was stopped by the addition of 2 × SDS sample buffer and proteins were separated on 10% SDS–PAGE gels.

**Statistical analysis.** Significance of differences in cell recovery, SG numbers, cell numbers and germ cell counting results was assessed by the two-tailed *t*-test.

**Data availability.** Data supporting the findings of this study are available within the article and its Supplementary Information; all data supporting findings are available on reasonable request.

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

## Acknowledgements

Z.Z. was supported by the National Institute of Genetics as an NIG postdoctoral fellow in 2012 and 2013. This work was supported by the Japan Society for the Promotion of Science KAKENHI in a grant (No. 26251025) to Y.S., and a Grant-in-Aid for Scientific Research on Innovative Areas ('Epigenome dynamics and regulation in germ cells'; No. 25112002) from the Ministry of Education, Culture, Sports, Science and Technology, Japan to Y.S.

## Author contributions

Z.Z. designed the experiments and performed data analysis. H.K. established the *Nedd4-cKO* mouse line and performed IP experiment in *Nedd4 KO* MEF cells. A.S. and K.S. conducted MAS analysis of NANOS2 complex. Z.Z., H.K. and Y.S. wrote the manuscript.

## Additional information

**Competing interests:** The authors declare no competing financial interests.

