## [Peer Review File · Nature Communications]

Reviewers' Comments:

Reviewer #1 (Remarks to the Author)

Ref: Nature Communication_ NCOMMS-16-20070

COMMENTS

The manuscript entitled "The Nedd4 Complex Controls Spermatogonial Stem Cell Homeostasis and Stress Response by Regulating the Dynamics of Messenger Ribonucleoprotein Complexes" by Zhi Z et al., could be an important finding in understanding the importance of NEDD4 in SG clearance although some concerns remain to be considered critically before taking any further decision. However, a number of concerns remain that precludes it to be considered for publication in the journal. Following are the excerpts:

1. Authors should justify the setting of two temperatures (viz. 33 and 42 deg) ---- is it physiological point-of-view or arbitrarily chosen?
2. Figure1B is not very clear and the distinction between two temperatures for the expression/localization pattern is not clearly defined in the figure.
3. Fig1F does not show the co-localization properly. Captured images indicate reduced expression of Nedd4 along with others at 42 degree temp. rather than their localization, otherwise not capturing the images with ideal proper exposure times for DAPI and others. Explain and should repeat the experiment to clear any doubt. What is tg in Supplementary Fig.1 A.
4. Co-IP in multiple figures (e.g., 1C&E) are not clear to make any distinct comment; need reverse Co-IP experiments to solidify the data.
5. Authors mention that NEDD4 cKD GSCs showed a lower rate of cell recovery which can be attributed to either (i) reduced proliferation or (ii) increased differentiation. However, upon qPCR analysis the authors observe increased expression of cell renewal markers and reduced expression of differentiation markers. This is a little confusing and needs some explanation.
6. In Fig 2B the effect of shNedd4 on PTEN is not distinct.
7. Authors claimed that Nanos2 can suppress both proliferation and differentiation by regulating fate of mRNAs. Why they think that the increased number of Nanos2 foci in NEDD4 cKD GSCs show preferentially more proliferation but not differentiation. A short introduction of Nanos2 would be useful to understand the results.
8. RA induced NDFIP2 expression while NEDD4 expression was unchanged. At this point in the manuscript, based only on these results it might not be enough to claim that NEDD4-NDFIP2 complex destabilizes Nanos2. However, this statement can be used later when authors show increased ubiquitination of Nanos2 upon co-expression of NEDD4 and NDFIP2.
9. Nanos2+ cells constitute 100% of SPC pool in NEDD4 cKO mice. What is the scaling of the graph in Fig 4E?
10. The authors have convincingly demonstrated how NEDD4 depletion can bear upon the stability of spermatogonial stem cells under heat stress. They further demonstrate how Nanos2 protein, as a component of mRNP complex under stress condition, is regulated by NEDD4 E3 ligase through endosomal pathway. However, the study, is somewhat correlative in the sense that there is no evidence to prove that the effects of NEDD4 are channeled through Nanos2. For example, a double knockdown of both NEDD4 and Nanos2 in GSCs?
11. In Fig7B the GST pull down assay shows the GST NEDD4 band which is very faint.
12. Also, there are concerns with grammatical errors and typos which make it difficult to get the meaning of quite a few sentences. The manuscript should be rearranged accordingly to eliminate any such error.

Reviewer #2 (Remarks to the Author)

This study from Dr Zhi and colleagues defines the role of NEDD4 in spermatogonia and specifically in its interaction with NANOS2 and mRNP complexes. This interaction is of importance in both normal spermatogenesis, in the commitment of spermatogonia to differentiation, and under

situations of stress - I do however, have difficulty with the authors attempts to translate this information to the human situation. The quality of the data is generally excellent and well controlled (noting a few points to address below). The data appear to be novel and it will likely make a valuable contribution to the field of spermatogonial stem cell biology specifically, and stem cell biology broadly.

Addressing the questions below will improve the quality of the manuscript. They are not in order of importance.

1. Are the Nedd4 germ cell specific KOs infertile? This is essential information.
2. What is the function of NEDD4 in spermatids as indicated by immunolabelling shown in Fig 1B and S1B?
3. Introduction. Varicocele and cryptorchidism are certainly not 'the' leading causes of human infertility. The majority of cryptorchidism is surgically corrected before the age of one with excellent results and the role of varicocele in infertility is highly controversial. The consensus appears to be that it is a risk factor. In relation to the later, the authors seem to have missed that varicocele is a disorder of the excurrent ductal system, rather than the testis per se, and the proposed mechanism is via reactive oxygen damage of sperm during transit from the testis. In extreme circumstances this may impact testis blood flow, but this is rare. My recommendation is that the authors remove this speculation. Excellent germ cell biology is reason enough to do and publish the study. Where this work may be relevant is in areas of the world where there are extreme summer temperatures. While I have not read this literature in detail for a long time, my recollection is that in several middle eastern countries, fertility rates decrease in summer.
4. line 66-7. Spermatogonia are 'near' rather than 'on' the surface of the seminiferous tubule ie. they are covered by basement membrane and peritubular cells.
5. Line 110. What does the term '.. even after active transcription' mean?
6. line 221. Neither Fig 3A or S3A show testis weight data - this should be shown.
7. lines 223-5. Presumably the authors are saying that the ratio of Sertoli cells to germ cells is increased in KO testis ie. germ cells are disappearing. More detail is required here. What are the histological defects. Fig 3B needs to be replaced with higher power and resolution images. Even following an examination of the submitted images I was unable to pin point which cells were abnormal. The authors need to define what is measured in Fig 3C and show appropriate images.
8. Fig 3B does however show that NEDD4 is not required for full spermatogenesis i.e. all germ cell types are present in the KO. It is required for quantitatively normal spermatogenesis. As such, the titles of S3 and S4 need to be corrected.
9. The scale bar in Fig 1B is incorrect ie. it is not 50um.
10. Fig S1B legend - should read 'whole' rather than 'while'.

Reviewer #3 (Remarks to the Author)

In this manuscript the authors provide data suggesting that Nedd4 targets Nanos2 in spermatogonial stem cells which leads to cell differentiation. They further show that Nedd4 may be required for the clearance of stress granules. While the observations are interesting, there are a number of significant issues that require consideration.

Major concerns

It is unclear what "control" mice were used in the study. If these were Cre-negative mice treated with tamoxifen, or non-tamoxifen treated males. As tamoxifen itself can have an effect on male mice (e.g. Taguchi, Biol Reprod 1987) it is important to include this information.

The red fluorescence is very difficult to see in most figures and in general panels too small to clearly evaluate the data presented. For example it is difficult to see SGs in most panels in Figure 1. Also it is unclear why Nedd4 staining is reduced in 1D and 1F at 42 C.

Knockdown experiments should include a tamoxifen-treated scrambled siRNA control.

What method was used to determine cell numbers for Figure 2D? Were dead cells excluded? What is meant by “cell recovery”? It would be good to use a method that only counts live cells such as MTT. Tamoxifen treatment might increase cell death in these cells, a possibility that has not been excluded in the data presented.

In Figure 2A, are these transcripts associated with self-renewal (Nanos2 and Gfra1) and differentiation (Sohlh1, Sohlh2, and Taf7l) really up- and down-regulated, respectively, or are there just more and less cells respectively, that express these genes (i.e. the ratio of positive to negative cells changed). This should be discussed.

In Figure S2D blot the tamoxifen treatment increased Nanos2 protein levels at 6 and 12 h, but this is not reflected in the graph in E. To really show that degradation is slowed down, the half-life of the protein should be presented in each case (and not levels shown as percentage of the starting levels).

The histological analysis mentioned in lines 216/217 is not shown and should be included.

The reduction of meiotic germ cells in Figure S3G is not very obvious. Some of the staining in the control, especially in the two tubules on the left of the image seem to be background, as the green fluorescence is right at the outside of the tubule, where no meiotic germ cells are expected. A higher magnification image may help.

It is unclear what the authors mean by “stage-dependent activation of p-RPS6 in both somatic and CDH+ spermatogonia” (lines 258/259)? Perhaps they mean “active p-RSP6 in somatic cells and stage-specific in spermatogonia” – either way, this is not apparent in the Figure S4C.

With the quality of figures provided it is difficult to see anything in the images in Figure 4. In addition, for Figure 4F, if the authors claim “even in RA-rich stages VII-XI of tubules” (line 265), it should be indicated in the Figure, which of the tubules were at these stages.

In Figure S5A there appear to be significant SGs in cells that have been treated with DMSO. Also, as the images shown in Figure S5A and Figure 5A look very similar, it is not possible to claim that “SGs with Nanos2 were disassembled in control GSCs” (line 280).

In data presented in Figure 6, 3-4 month old mice were used. At this age, the mice already display a phenotype (Figure 3) – how does this affect the results? This should be discussed.

In Figure 6C that there are many more PARP-positive (green) cells in the control compared to the knockout. How can this be explained?

Ndfip1 and Ndfip2 are strong binders of Nedd4 WW domains (that is how they were discovered), so it is unsurprising that they interact and coprecipitate from cells overexpressing Nedd4. In Figure 7B the authors show that purified recombinant Nedd4 and Nanos2 directly interact without adaptor proteins like Ndfip1 and Ndfip2, and in 7H they show that Nedd4 can ubiquitinate Nanos2 in reconstituted *in vitro* ubiquitination assays. Thus the *in vivo* role of Ndfip2 in Nanos1 regulation is confusing and unclear. Furthermore, Ndfip2 KO mice are not known for any defects in spermatogenesis.

To confirm that if Nanos1 is a target of Nedd4 *in vivo*, the authors should provide evidence that ubiquitination of Nanos2 is abrogated in Nedd4 cKO cells.

Loss of Nedd4 results in severe growth retardation due to reduced IGF1 and insulin signalling. Even the loss of a single Nedd4 allele in mouse results in reduced animal and organ size. Did the author consider if reduced growth signalling through IGF1 and insulin contribute to the phenotype

in their cKO?

Other Comments

Scale bars are missing in higher magnification in Figure 1.

The images in Figure S3 are very low resolution, making it very difficult to see the loss of Nedd4 in S3C.

Line 193, 198, the knockdown of Nedd4 does not result in a complete absence of Nedd4 – this needs to be re-worded.

Gene and protein symbol nomenclature is inconsistent. It is recommended to use the nomenclature as outlined on the Jackson laboratory webpage.

All Figure legends should state how many independent biological replicates were performed if statistical analysis is indicated.

Many original citations are missing.

Reviewer #4 (Remarks to the Author)

The work by Zhou Zhi et al. provides new insights into biology of RNA metabolism and RNA granules and their roles in spermatogonial stem cell homeostasis. However, I cannot recommend it for publication in the current state.

1) There is a consistent problem with using names SGs, P bodies and mRNP complexes interchangeably. The authors have not demonstrated that Nanos2/Nedd4 complexes are mRNPs.

2) The discussion part is too long and speculative

3) Figure 1. Is Nedd4/Nanos2 interaction RNA-dependent? What is an effect of Nanos2 depletion/overexpression on SG formation?

4) Not every stress-induced cytoplasmic foci are SGs (even in the presence of SG marker). Authors must use FISH to demonstrate that heat-induced Nedd4/Nanos2-positive foci contain polyadenylated mRNAs. Do they contain 40S ribosomal subunits and translation initiation factors (such as eIF3, eIF4G)?

5) Figure 1. Figures 1D and 1F are of poor quality. What is the percentage of cells forming SGs?

6) Figure 2. Line 766: What is (D)?

In Figure 2D (cell number at 5-day intervals), what is expression of differentiation-related and spermatogonial self-renewal transcripts? It makes more sense to look at them at 15 days point rather than at 96 hours (Figure 2A)

Figure 2H and 2I: treatment with cycloheximide is extremely long (up to 9 hours)

7) Is Nedd4 and/or Nanos2 are part of SGs caused by other stresses (e.g. sodium arsenite). Are sodium arsenite-induced SGs also cleared through endosomal/lysosomal pathway?

8) Figure 8 states (A). Is there also other sub-figures? The legends to this figure have no explanation to the model provided

Minor points:

I.52 "P bodies usually consist....with translational repressors". It is not entirely true for P-bodies (Only Ago proteins are translational repressors found in PBs), yet correct for SGs.

I.57 "Hypoxia and low nutrient supply". These are poor conditions to stimulate SG formation when compared to oxidative stress or UV light

Scale bars have to be moved into figures.

Reviewers' comments:

Reviewer #1 (Remarks to the Author):

Ref: Nature Communication_ NCOMMS-16-20070

COMMENTS

The manuscript entitled "The Nedd4 Complex Controls Spermatogonial Stem Cell Homeostasis and Stress Response by Regulating the Dynamics of Messenger Ribonucleoprotein Complexes" by Zhi Z et al., could be an important finding in understanding the importance of NEDD4 in SG clearance although some concerns remain to be considered critically before taking any further decision. However, a number of concerns remain that precludes it to be considered for publication in the journal. Following are the excerpts:

1. Authors should justify the setting of two temperatures (viz. 33 and 42 deg) ---- is it physiological point-of-view or arbitrarily chosen?

R: We appreciate the reviewer's question. Temperatures in this study are based on both the literature and our preliminary experiments. Given that the testicular temperature is maintained at 2°C–4°C below the core body temperature (37°C)(Kim et al., 2012; Mieusset and Bujan, 1995), we set 33°C as the starting temperature. We further used normal body temperature, 37°C, and a heat stress temperature, 42°C, used in a previous study (Kim et al., 2012). By checking the stress granule formation using different temperatures, we observed a significant induction of stress granules (EIF4E staining) in cultured GSCs at 42°C but not at the lower temperatures, 37°C and 33°C (Figure Q1), which is consistent with previous observations (Kim et al., 2012). Thus, we showed the results of two temperatures (33 °C for normal testicular temperature and 42°C for heat stress testicular temperature) in our study.

Figure Q1: Rapid SG formation at 42°C but not 33 °C or 37°C.

2. Figure1B is not very clear and the distinction between two temperatures for the expression/localization pattern is not clearly defined in the figure.

R: We thank the reviewer for commenting on this important point. We are afraid that the poor resolution is partly caused by the compression to PDF. We tried to indicate cells forming SGs at 42°C in Figure 1B in the revised manuscript.

3. Fig1F does not show the co-localization properly. Captured images indicate reduced

expression of Nedd4 along with others at 42 degree temp. rather than their localization, otherwise not capturing the images with ideal proper exposure times for DAPI and others. Explain and should repeat the experiment to clear any doubt. What is tg in Supplementary Fig.1 A.

R: We thank the reviewer for their comments. We did not observe a quantitative Nedd4 protein level change after heat stress for 20 min as shown by WB in figure Q2 A. We have shown other clones in Figure Q2 B. Tg is a mouse line that expresses Flag-Tagged Nanos2 under the control of the *Nanos2* enhancer established before (Suzuki et al. 2010, 2012). This has been stated in the revised manuscript.

Figure Q2: (A) Western blotting results from GSCs cultured at the indicated temperature for 20 min. (B) Immunostaining of GSCs cultured at 33 and 42 degrees with Nedd4 (green) and Dazl (magenta) antibodies.

4. Co-IP in multiple figures (e.g., 1C&E) are not clear to make any distinct comment; need reverse Co-IP experiments to solidify the data.

R: We are sorry if we confused the reviewer. We showed Fig. 1E as a reverse Co-IP for 1C. Now, we provide additional data for the reverse Co-IP as shown in Figure Q3 as Figure S1H (related to Figure 1E) in the revised manuscript. It is clear that Nanos2 and Nedd4 interact with each other in GSCs.

Figure Q3: Western blots of a co-IP experiment with an anti-Nanos2 antibody using protein lysates from GSCs incubated at 33°C (open circles) and 42°C (filled circles) for 20 min.

5. Authors mention that NEDD4 cKD GSCs showed a lower rate of cell recovery, which can be attributed to either (i) reduced proliferation or (ii) increased differentiation. However, upon qPCR analysis the authors observe increased expression of cell renewal markers and reduced expression of differentiation markers. This is a little confusing and needs some explanation.

R: We appreciate the reviewer's question. In our study, we would like to show the reduced cell growth in NEDD4 cKD GSCs. In these cells, we observed increased

expression of cell renewal markers and reduced expression of differentiation markers. We have changed our description to avoid any confusion.

6. In Fig 2B the effect of shNedd4 on PTEN is not distinct.

R: We are not sure about the reviewer's question. As Fig2C indicated, we also think that the PTEN level is not changed by knocking down Nedd4.

7. Authors claimed that Nanos2 can suppress both proliferation and differentiation by regulating fate of mRNAs. Why they think that the increased number of Nanos2 foci in NEDD4 cKD GSCs show preferentially more proliferation but not differentiation. A short introduction of Nanos2 would be useful to understand the results.

R: We appreciate the reviewer's question. Actually, we think that Nedd4 cKD causes decreased proliferation as well as differentiation. We are sorry for the confusing description and we have changed the description in the revised manuscript.

8. RA induced NDFIP2 expression while NEDD4 expression was unchanged. At this point in the manuscript, based only on these results it might not be enough to claim that NEDD4-NDFIP2 complex destabilizes Nanos2. However, this statement can be used later when authors show increased ubiquitination of Nanos2 upon co-expression of NEDD4 and NDFIP2.

R: We agree with the reviewer's opinion. We have changed the description in the revised manuscript.

9. Nanos2+ cells constitute 100% of SPC pool in NEDD4 cKO mice. What is the scaling of the graph in Fig 4E?

R: In this part, we compared the proportion of Nanos2/CDH1-double-positive cells in CDH1-positive cells between control and Nedd4 cKO testes.

10. The authors have convincingly demonstrated how NEDD4 depletion can bear upon the stability of spermatogonial stem cells under heat stress. They further demonstrate how Nanos2 protein, as a component of mRNP complex under stress condition, is regulated by NEDD4 E3 ligase through endosomal pathway. However, the study, is somewhat correlative in the sense that there is no evidence to prove that the effects of NEDD4 are channeled through Nanos2. For example, a double knockdown of both NEDD4 and Nanos2 in GSCs?

R: We appreciate the reviewer's comments. As the reviewer's suggested, we tested Nanos2 knockdown by several lentivirus shRNAs targeting the *Nanos2* gene; however, Nanos2 is a one-exon gene and none of these shRNAs effectively decreased the Nanos2 level (data not shown). We therefore conducted overexpression of Nedd4 and Nanos2 using a lentivirus system. As shown in the following figures, overexpression of Nedd4 reduced Nanos2 protein (Figure Q4, A) and also lead to an increase of the differentiation transcripts *Sohlh1*, *Sohlh2* and *Taf7l* (Figure Q4, B). However, further overexpression of Nanos2 significantly decreased these differentiation genes (Figure Q4, B). These data

suggested that the Nedd4 effects in GSCs were partially through Nanos2. We added this data in the revised manuscript in Figure S2D and E.

Figure Q4: (A) Western blot analysis of GSCs infected with the indicated lentivirus vectors for 96 hours. (B) qPCR analysis of the key spermatogonial differentiation-related genes with the indicated treatment. (* $p < 0.05$; ** $p < 0.01$).

11. In Fig7B the GST pull down assay shows the GST NEDD4 band which is very faint.
 R: We appreciate the reviewer's comment. We have shown the blotting of Nanos2 but not Nedd4. In the following figure, we show the GST-Nedd4 staining in the right panel (Figure Q5).

Figure Q5: Western blot analysis of MBP-Nanos2 purified with either GST or GST-Nedd4. Yellow arrowheads indicate nonspecific bands. The leftmost panel indicates Nedd4 staining.

12. Also, there are concerns with grammatical errors and typos which make it difficult to get the meaning of quite a few sentences. The manuscript should be rearranged accordingly to eliminate any such error.

R: We apologize for these mistakes and we have received English editing.

Reviewer #2 (Remarks to the Author):

This study from Dr Zhi and colleagues defines the role of NEDD4 in spermatogonia and specifically in its interaction with NANOS2 and mRNP complexes. This interaction is of importance in both normal spermatogenesis, in the commitment of spermatogonia to differentiation, and under situations of stress - I do however, have difficulty with the authors attempts to translate this information to the human situation. The quality of the data is generally excellent and well controlled (noting a few points to address below). The data appear to be novel and it will likely make a valuable contribution to the field of spermatogonial stem cell biology specifically, and stem cell biology broadly.

Addressing the questions below will improve the quality of the manuscript. They are not in order of importance.

1. Are the Nedd4 germ cell specific KOs infertile? This is essential information.

R: We appreciate the reviewer's question. We described the fact that these mice are fertile until a least 6 months in the revised manuscript.

2. What is the function of NEDD4 in spermatids as indicated by immunolabelling shown in Fig 1B and S1B?

R: We appreciate the reviewer's question. In the current manuscript, we focused on the function of NEDD4 in the spermatogonial population. As for the function in spermatids, we do not yet have data to predict this nor do we have a suitable Cre system to delete Nedd4 specifically in spermatids. Of course we are planning to set up a system for this. However, we feel that this is beyond the scope of the current manuscript.

3. Introduction. Varicocele and cryptorchidisms are certainly not 'the' leading causes of human infertility. The majority of cryptorchidism is surgically corrected before the age of one with excellent results and the role of varicocele in infertility is highly controversial. The consensus appears to be that it is a risk factor. In relation to the later, the authors seem to have missed that varicocele is a disorder of the excurrent ductal system, rather than the testis per se, and the proposed mechanism is via reactive oxygen damage of sperm during transit from the testis. In extreme circumstances this may impact testis blood flow, but this is rare. My recommendation is that the authors remove this speculation. Excellent germ cell biology is reason enough to do and publish the study. Where this work may be relevant is in areas of the world where there are extreme summer temperatures. While I have not read this literature in detail for a long time, my recollection is that in several middle eastern countries, fertility rates decrease in summer.

R: We appreciate the reviewer's comments. We have deleted the description regarding human infertility in our introduction and discussion.

4. line 66-7. Spermatogonia are 'near' rather than 'on' the surface of the seminiferous tubule ie. they are covered by basement membrane and peritubular cells.

R: Corrected.

5. Line 110. What does the term '.. even after active transcription' mean?

R: In our previous study, we found that embryonic *Nanos2* mRNA transcription is only active during E13.5 and E14.5, and becomes significantly reduced from E15.5, but the *Nanos2* protein level was stable even in the neonatal stage. However, the protein has to be cleared before differentiation. This is why we are interested in the regulatory mechanism of *Nanos2* protein. We deleted this confusing description and rewrote our rationale in the revised manuscript.

6. line 221. Neither Fig 3A or S3A show testis weight data - this should be shown.

R: We appreciate the reviewer's comments. We have shown testis weight data in Figure 3A in the revised manuscript related to Figure 3B.

7. lines 223-5. Presumably the authors are saying that the ratio of Sertoli cells to germ cells is increased in KO testis ie. germ cells are disappearing. More detail is required here. What are the histological defects. Fig 3B needs to be replaced with higher power and resolution images. Even following an examination of the submitted images I was unable to pin point which cells were abnormal. The authors need to define what is measured in Fig 3C and show appropriate images.

R: We appreciate the reviewer's comments. We added higher resolution pictures in Figure 3C. We counted tubules with significantly fewer germ cells as abnormal testicular tubules and we have defined these criteria in the revised manuscript.

8. Fig 3B does however show that NEDD4 is not required for full spermatogenesis i.e. all germ cell types are present in the KO. It is required for quantitatively normal spermatogenesis. As such, the titles of S3 and S4 need to be corrected.

R: We appreciate the reviewer's comments. As the reviewer pointed out and correctly described, *Nedd4* KO testes contain all types of spermatocytes, even though the numbers are reduced and some of them exhibit abnormal morphology. Thus, we have changed our titles of S3 and S4 in the revised manuscript.

9. The scale bar in Fig 1B is incorrect ie. it is not 50um.

R: Corrected.

10. Fig S1B legend - should read 'whole' rather than 'while'.

R: Corrected.

Reviewer #3 (Remarks to the Author):

In this manuscript the authors provide data suggesting that *Nedd4* targets *Nanos2* in spermatogonial stem cells which leads to cell differentiation. They further show that

Nedd4 may be required for the clearance of stress granules. While the observations are interesting, there are a number of significant issues that require consideration.

Major concerns

It is unclear what “control” mice were used in the study. If these were Cre-negative mice treated with tamoxifen, or non-tamoxifen treated males. As tamoxifen itself can have an effect on male mice (e.g. Taguchi, Biol Reprod 1987) it is important to include this information.

R: We appreciate the reviewer’s comments. We did not treat the mice with tamoxifen in the mice study in the current manuscript because the Nanos3 Cre is not inducible. The “control” mice are Nanos3 Cre-positive Nedd4 *f*+ mice, while experimental mice are Nanos3 Cre-positive Nedd4 *f*/Δ mice.

The red fluorescence is very difficult to see in most figures and in general panels too small to clearly evaluate the data presented. For example it is difficult to see SGs in most panels in Figure 1. Also it is unclear why Nedd4 staining is reduced in 1D and 1F at 42 C.

R: We are sorry for the difficulty. The compression of the initial submission into a PDF may have caused this problem and we have improved this by sending less compressed files. As for the staining of Nedd4, we have quantitative data by WB, and demonstrated no change in the Nedd4 protein level during the 20-min heat stress.

Knockdown experiments should include a tamoxifen-treated scrambled siRNA control.

R: We appreciate the reviewer’s comments. We examined scrambled shRNA-infected cells treated with DMSO or tamoxifen. The results showed no effects on Nedd4 expression (Figure Q6).

Figure Q6: (A) Western blot analysis of GSCs infected with scrambled-shRNA lentivirus for 96 hours. (B) qPCR analysis of *Nedd4* and other key spermatogonial-related genes with the indicated treatment. (* $p < 0.05$; ** $p < 0.01$).

What method was used to determine cell numbers for Figure 2D? Were dead cells excluded? What is meant by “cell recovery”? It would be good to use a method that only counts live cells such as MTT. Tamoxifen treatment might increase cell death in these cells, a possibility that has not been excluded in the data presented.

R: The cell recovery experiment was performed according to previous reports. GSC clones were gently collected and trypsinized as single cell suspension, trypan blue staining was performed before cell counting to exclude dead cells. Regarding the Tamoxifen treatment, we used a lentivirus-mediated stable knockdown system and the

Cre efficiency was monitored by GFP (Figure S3, C). Tamoxifen was removed once the GFP signal was lost (after 72 hours), thus the tamoxifen would not have a long-term effect on cells (15 days counting). During the 72-hour treatment, we performed apoptotic cell assays, and no significant apoptosis was observed. As the GSCs are grown on feeder cells that may cause background in the MTT assay, we did not choose this method.

In Figure 2A, are these transcripts associated with self-renewal (Nanos2 and Gfra1) and differentiation (Sohlh1, Sohlh2, and Taf7l) really up-and down-regulated, respectively, or are there just more and less cells respectively, that express these genes (i.e. the ratio of positive to negative cells changed). This should be discussed.

R: We appreciate the reviewer's comments. We used a lentivirus-mediated stable knockdown system for our GSC system; the Cre efficiency was monitored by GFP (Figure S3, C). Staining of Nedd4 showed that more than 95% cells were successfully knocked down. Therefore, we think that the up-and down-regulation was not caused by a change in the ratio of positive to negative cells.

In Figure S2D blot the tamoxifen treatment increased Nanos2 protein levels at 6 and 12 h, but this is not reflected in the graph in E. To really show that degradation is slowed down, the half-life of the protein should be presented in each case (and not levels shown as percentage of the starting levels).

R: We are sorry for the misleading description. We do not claim that tamoxifen treatment increased Nanos2 protein levels at 6 and 12 h, rather we want to show that Nanos2 repression by RA treatment was blocked by Nedd4 depletion in this figure. Graph E reflects the relative Nanos2 level (normalized with β -tubulin), thus we think it is reasonable to show this by comparing with the initial amount.

The histological analysis mentioned in lines 216/217 is not shown and should be included.

R: According to the reviewer's comment, we added the histological analysis in Figure S3C in the revised manuscript.

The reduction of meiotic germ cells in Figure S3G is not very obvious. Some of the staining in the control, especially in the two tubules on the left of the image seem to be background, as the green fluorescence is right at the outside of the tubule, where no meiotic germ cells are expected. A higher magnification image may help.

R: We appreciate the reviewer's comment. As shown in the higher magnification image below, green fluorescence is not the outside of the tubule, but is from the second layer, and the staining pattern indicates the chromosomes of meiotic cells. Thus, we regard these as positive cells (Figure Q7). DAPI in blue is difficult to see in low magnification. This may be why the reviewer had difficulty in judging from our original results. We apologize for this and have used a higher magnification in the new Figure S3F.

Figure Q7: Representative staining shows that the SYCP3⁺ meiotic cells were reduced in Nedd4 cKO testis as compared with the control. Asterisks indicate defective tubules. Magnified pictures are shown in the lowest panel.

It is unclear what the authors mean by “stage-dependent activation of p-RPS6 in both somatic and CDH⁺ spermatogonia” (lines 258/259)? Perhaps they mean “active p-RPS6 in somatic cells and stage-specific in spermatogonia” – either way, this is not apparent in the Figure S4C.

R: We thank the reviewer for this comment. Our intended meaning was that active p-RPS6 signals are observed in both somatic cells and spermatogonia in a stage dependent manner. It is already reported that stage-dependent activation of p-RPS6 is observed during spermatogenesis (Hobbs et al., 2015), and we also show this in figure 4F (p-RPS6, green) and S4D. In figure S4C, we only used a magnified picture showing the activation of p-RPS6 in CDH1⁺ cells at the indicated stages.

With the quality of figures provided it is difficult to see anything in the images in Figure 4. In addition, for Figure 4F, if the authors claim “even in RA-rich stages VII-XI of tubules” (line 265), it should be indicated in the Figure, which of the tubules were at these stages.

R: We appreciate the reviewer’s comment. The major reason of the low quality is due to the compression of the files for initial submission. We believe that our original pictures have higher quality. Nevertheless, we have added the indications as the reviewer suggested in the figure legends.

In Figure S5A there appear to be significant SGs in cells that have been treated with DMSO. Also, as the images shown in Figure S5A and Figure 5A look very similar, it is not possible to claim that “SGs with Nanos2 were disassembled in control GSCs” (line 280).

R: We apologize for some of our data being complicated. In Figure S5A, SGs were formed by heat shock (42°C) treatment but not with DMSO. Through these results, we would like to show the co-localization of Nanos2 and TfR after heat shock. In Figure 5B, the temperature was shifted from 42°C to 33°C, and the SGs were disassembled in control (DMSO) but not in Nedd4-KD (4-OHT) cells. We think that the lack of disassembly in Nedd4-KD is obvious. We have changed our misleading description as well.

In data presented in Figure 6, 3-4 month old mice were used. At this age, the mice already display a phenotype (Figure 3) – how does this affect the results? This should be discussed.

R: We understand the reviewer’s question. As the reviewer pointed out, abnormalities already exist in Nedd4-cKO mice. However, we focused on the regeneration defects of spermatogonia after heat stress between Nedd4 cKO mice and control mice, and we would like to show that the defects are accelerated by heat stress in Nedd4 cKO mice (Figure S6 G). We also compared littermate Nedd4 cKO mice treated with or without heat stress, and results indicated that more severe defects occurred in Nedd4 cKO mice by heat stress (Figure Q8). We discuss this in the revised manuscript.

Figure Q8: A, Testis slices from Nedd4 cKO mice treated with or without heat stress were examined after hematoxylin and eosin (H&E) staining. B, abnormal testicular tubules with significantly reduced germ cells (indicated by asterisks) were counted and shown as mean ± SD. Scale bars = 50 μm.

In Figure 6C that there are many more PARP-positive (green) cells in the control compared to the knockout. How can this be explained?

R: We appreciate the reviewer's question. Seven-day heat stress induced apoptosis in most meiotic cells in both control and knockout mice. At this stage (3-4 months), meiotic cells were significantly decreased in knockout animals, and we think this is the reason why more apoptotic cells were observed in control mice.

Ndfip1 and Ndfip2 are strong binders of Nedd4 WW domains (that is how they were discovered), so it is unsurprising that they interact and coprecipitate from cells overexpressing Nedd4. In Figure 7B the authors show that purified recombinant Nedd4 and Nanos2 directly interact without adaptor proteins like Ndfip1 and Ndfip2, and in 7H they show that Nedd4 can ubiquitinate Nanos2 in reconstituted in vitro ubiquitination assays. Thus the in vivo role of Ndfip2 in Nanos1 regulation is confusing and unclear. Furthermore, Ndfip2 KO mice are not known for any defects in spermatogenesis. To confirm that if Nanos1 is a target of Nedd4 in vivo, the authors should provide evidence that ubiquitination of Nanos2 is abrogated in Nedd4 cKO cells.

R: We appreciate the reviewer's question. As the reviewer suggested, we checked whether endogenous Nedd4 regulated Nanos2 protein (Figure Q9). For this purpose, we used Nedd4 KO MEF cells. Myc-tagged Nanos2 was transfected into both WT and Nedd4 KO MEF cells. Anti-Myc-IP was performed followed by checking ubiquitination. In wild type MEF cells, ubiquitinated Nanos2 was observed; however, in the absence of Nedd4, Nanos2 ubiquitination is significantly abrogated. We added this information in the revised manuscript Figure S7C.

Figure Q9: Western blot analysis showed that loss of endogenous Nedd4 reduced Nanos2 protein ubiquitination.

Loss of Nedd4 results in severe growth retardation due to reduced IGF1 and insulin signalling. Even the loss of a single Nedd4 allele in mouse results in reduced animal and organ size. Did the author consider if reduced growth signalling through IGF1 and insulin contribute to the phenotype in their cKO?

R: We thank the reviewer for their comments. We have examined the reported Nedd4 target IGF-1R (Kwak et al., 2012) in our Nedd4 cKD GSCs, but no significant changes were observed. In addition, the major downstream targets of IGF-1 signaling in SSCs (Wang et al., 2015), pAKT and pERK, were also not changed in the absence of Nedd4. Thus, we think that most Nedd4 function in GSCs is not through IGF1 signaling. However, we cannot fully exclude the possibility of IGF-1 signaling being regulated through Nedd4 in testes, and we will further examine this in the future.

Other Comments

Scale bars are missing in higher magnification in Figure 1.

R: Corrected.

The images in Figure S3 are very low resolution, making it very difficult to see the loss of Nedd4 in S3C.

R: Corrected.

Line 193, 198, the knockdown of Nedd4 does not result in a complete absence of Nedd4 – this needs to be re-worded.

R: We removed this section.

Gene and protein symbol nomenclature is inconsistent. It is recommended to use the nomenclature as outlined on the Jackson laboratory webpage.

R: We appreciate the reviewer's comments. We have changed these symbols.

All Figure legends should state how many independent biological replicates were performed if statistical analysis is indicated.

R: We have added sample information.

Many original citations are missing.

R: We now provide the original references.

Reviewer #4 (Remarks to the Author):

The work by Zhou Zhi et al. provides new insights into biology of RNA metabolism and RNA granules and their roles in spermatogonial stem cell homeostasis. However, I cannot recommend it for publication in the current state.

1) There is a consistent problem with using names SGs, P bodies and mRNP complexes interchangeably. The authors have not demonstrated that Nanos2/Nedd4 complexes are mRNPs.

R: We appreciate the reviewer's comments.

We apologize for calling Nanos2/Nedd4 a complex in Figure S7, but we do not claim it to be an mRNP. We have shown that Nanos2 is an mRNP component in GSCs (Zhou et al., 2015) and gonocytes (Suzuki et al., 2010). In the current manuscript, Nedd4 interacted with Nanos2 for targeting its degradation, but Nedd4 is not an mRNP component. We have changed our description in the revised manuscript.

2) The discussion part is too long and speculative

R: According to the reviewer's suggestion, we have revised our discussion.

3) Figure 1. Is Nedd4/Nanos2 interaction RNA-dependent? What is an effect of Nanos2 depletion/overexpression on SG formation?

R: We thank the reviewer for their questions. We checked Nedd4/Nanos2 interaction by treating the lysates with 1 mg/ml of RNAase A, and then performed immunoprecipitation. The results suggested that the Nedd4 and Nanos2 interaction is RNA-independent (Figure Q10). This result is shown as the new Fig. S1G. We did not examine Nanos2 effects in SG formation in the current manuscript because our previous study demonstrated that Nanos2 localizes to P-bodies and deletion of Nanos2 does not affect P-body formation itself (Suzuki et al., 2010).

Figure Q10: Western blot analysis showed that interaction of Nedd4 and Nanos2 was RNA independent.

4) Not every stress-induced cytoplasmic foci are SGs (even in the presence of SG marker). Authors must use FISH to demonstrate that heat-induced Nedd4/Nanos2-positive foci contain polyadenylated mRNAs. Do they contain 40S ribosomal subunits and translation initiation factors(such as eIF3, eIF4G)?

R: We appreciate the reviewer's comments. We tried to perform co-localization of Nedd4/Nanos2 with *polyA*⁺ mRNAs (Kim et al., 2012), but we failed to detect Nanos2/Nedd4 after FISH staining. This may be because the antibodies are not suitable for co-staining in FISH conditions. EIF4E is one of the translation initiation factors and we performed co-staining to show that some of the Nanos2 foci co-localized with EIF4E (Figure Q11).

Figure Q11: (A) immunostaining showed co-localization of the SG marker EIF4E with Nanos2.

5) Figure 1. Figures 1D and 1F are of poor quality. What is the percentage of cells forming SGs?

R: We appreciate the reviewer's comment. The major reason for the low quality is the compression of files for initial submission. We believe that our original pictures are higher quality. The percentage of cells with SGs after heat stress was $91.7 \pm 3.7\%$.

6) Figure 2. Line 766: What is (D)?

In Figure 2D (cell number at 5-day intervals), what is expression of differentiation-related and spermatogonial self-renewal transcripts? It makes more sense to look at them at 15 days point rather than at 96 hours (Figure 2A)

Figure 2H and 2I: treatment with cycloheximide is extremely long (up to 9 hours)

R: We are sorry for the typo. It should be (C) in line 766, not (D). We have changed this in the revised manuscript. In Figure 2D, we checked the cell number in the Nedd4 cKD stable line treated with DMSO or 4-OHT. As shown in Figure S2B and C, after 4-OHT treatment for 96 hours, GFP was completely lost, indicating establishment of a stable knockdown cell line by 4-OHT treatment. Thus, it is reasonable to check self-renewal and differentiation transcript expression just after Nedd4 knockdown in Figure 2A. As for figure 2D, we only showed growth of these stable lines, and we do not think gene expression changed much in the stable lines.

In Figure 2H and 2I, we chose these time points because Nanos2 protein was quite stable in male embryonic gonads (Sada et al., 2009; Tsuda et al., 2003) and also because of the slow cell growth properties of GSCs. We have preliminary experiments using short time points (up to 3 hours) as shown in the following figure; we observed Nanos2 protein degradation starting from 2 to 3 hours (Figure Q12). Thus, we used the time points shown in the manuscript.

Figure Q12: Western blot analysis of GSCs incubated with CHX (20 nM) for the indicated periods. The cells were harvested for western blotting to detect the Nanos2 protein.

7) Is Nedd4 and/or Nanog2 are part of SGs caused by other stresses (e.g. sodium arsenite). Are sodium arsenite-induced SGs also cleared through endosomal/lysosomal pathway?

R: We appreciate the reviewer's comments. We used heat stress to monitor the thermo-tolerance of germ cells, which happened in physiological conditions. As the reviewer suggested, sodium arsenite is also an inducer of SG formation, but is not in the physiological conditions in the testis. We are not excluding the possibility of the involvement of other physiological stresses, such as hypoxia or low nutrient supply, that also induce SGs; however, we feel analysis of those stresses is beyond the scope of the current manuscript.

8) Figure 8 states (A). Is there also other sub-figures? The legends to this figure have no explanation to the model provided

R: We are sorry for this confusion. There are no sub-figures for Figure 8. We have revised the figure legend.

Minor points:

1.52 "P bodies usually consist...with translational repressors". It is not entirely true for P-bodies (Only Ago proteins are translational repressors found in PBs), yet correct for SGs.

R: Corrected.

1.57 "Hypoxia and low nutrient supply". These are poor conditions to stimulate SG formation when compared to oxidative stress or UV light

R: Corrected.

Scale bars have to be moved into figures.

R: Corrected.

Hobbs, R.M., La, H.M., Makela, J.A., Kobayashi, T., Noda, T., and Pandolfi, P.P. (2015). Distinct germline progenitor subsets defined through Tsc2-mTORC1 signaling. *EMBO reports* 16, 467-480.

Kim, B., Cooke, H.J., and Rhee, K. (2012). DAZL is essential for stress granule formation implicated in germ cell survival upon heat stress. *Development* 139, 568-578.

Kwak, Y.D., Wang, B., Li, J.J., Wang, R., Deng, Q., Diao, S., Chen, Y., Xu, R., Masliah, E., Xu, H., *et al.* (2012). Upregulation of the E3 ligase NEDD4-1 by oxidative stress degrades IGF-1 receptor protein in neurodegeneration. *The Journal of neuroscience : the official journal of the Society for Neuroscience* 32, 10971-10981.

Mieusset, R., and Bujan, L. (1995). Testicular heating and its possible contributions to male infertility: a review. *International journal of andrology* 18, 169-184.

Sada, A., Suzuki, A., Suzuki, H., and Saga, Y. (2009). The RNA-binding protein NANOS2 is required to maintain murine spermatogonial stem cells. *Science* 325, 1394-1398.

Suzuki, A., Igarashi, K., Aisaki, K., Kanno, J., and Saga, Y. (2010). NANOS2 interacts with the CCR4-NOT deadenylation complex and leads to suppression of specific RNAs. *Proceedings of the National Academy of Sciences of the United States of America* 107, 3594-3599.

Tsuda, M., Sasaoka, Y., Kiso, M., Abe, K., Haraguchi, S., Kobayashi, S., and Saga, Y. (2003). Conserved role of nanos proteins in germ cell development. *Science* 301, 1239-1241.

Wang, S., Wang, X., Wu, Y., and Han, C. (2015). IGF-1R signaling is essential for the proliferation of cultured mouse spermatogonial stem cells by promoting the G2/M progression of the cell cycle. *Stem cells and development* 24, 471-483.

Zhou, Z., Shirakawa, T., Ohbo, K., Sada, A., Wu, Q., Hasegawa, K., Saba, R., and Saga, Y. (2015). RNA Binding Protein Nanos2 Organizes Post-transcriptional Buffering System to Retain Primitive State of Mouse Spermatogonial Stem Cells. *Developmental cell* 34, 96-107.

Reviewers' Comments:

Reviewer #1:

Remarks to the Author:

No comments.

Reviewer #2:

Remarks to the Author:

I am satisfied with the edits made to this paper. It is significantly improved and will make a valuable contribution to the field.

Regards Moira O'Bryan

Reviewer #3:

Remarks to the Author:

Authors have addressed my main concerns in the revised version of their manuscript.

Reviewer #4:

Remarks to the Author:

All my requests were addressed adequately. I recommend the manuscript for publication with the only recommendation to proofread English.

REVIEWERS' COMMENTS:

Reviewer #1 (Remarks to the Author):

No comments.

R: We thank the reviewer's suggestions on this work.

--

Reviewer #2 (Remarks to the Author):

I am satisfied with the edits made to this paper. It is significantly improved and will make a valuable contribution to the field.

R: We thank the reviewer's suggestions on this work.

--

Reviewer #3 (Remarks to the Author):

Authors have addressed my main concerns in the revised version of their manuscript.

R: We thank the reviewer's suggestions on this work.

--

Reviewer #4 (Remarks to the Author):

All my requests were addressed adequately. I recommend the manuscript for publication with the only recommendation to proofread English.

R: We thank the reviewer's suggestions on this work, and we have already asked a native for English editing.